# A Multi-Epitope Recombinant Vaccine Candidate Against Bovine Alphaherpesvirus 1 and 5 Elicits Robust Immune Responses in Mice and Rabbits

**DOI:** 10.3390/vaccines13111115

**Published:** 2025-10-30

**Authors:** Aline Aparecida Silva Barbosa, Samille Henriques Pereira, Mateus Laguardia-Nascimento, Amanda Borges Ferrari, Laura Jorge Cox, Raissa Prado Rocha, Victor Augusto Teixeira Leocádio, Ágata Lopes Ribeiro, Karine Lima Lourenço, Flávio Guimarães Da Fonseca, Edel F. Barbosa-Stancioli

**Affiliations:** 1Laboratório de Virologia Básica e Aplicada (LVBA), Departamento de Microbiologia, Instituto de Ciências Biológicas, Universidade Federal de Minas Gerais, Belo Horizonte 31270-720, Brazilsamillehenriques@gmail.com (S.H.P.); mateuslaguardia@gmail.com (M.L.-N.); amanda231182@gmail.com (A.B.F.); laura.cox@outlook.com (L.J.C.); vtleocadio@gmail.com (V.A.T.L.); agata.lribeiro@gmail.com (Á.L.R.); fdafonseca@icb.ufmg.br (F.G.D.F.); 2School of Biosciences, University of Surrey, Guilford GU2 7XH, UK; raissa.biotec@gmail.com; 3Laboratório de Pesquisas em Virologia, Faculdade de Medicina de São José do Rio Preto, São José do Rio Preto 15090-000, Brazil; karine_lourenco@hotmail.com

**Keywords:** *Varicellovirus bovinealpha1*, *Varicellovirus bovinealpha5*, BoAHV-1, BoAHV-5, recombinant vaccine, multi-epitope antigen, MVA vector

## Abstract

Background/Objectives: *Varicellovirus bovinealpha1* and *Varicellovirus bovinealpha5* (BoAHV-1 and BoAHV-5), respectively, are widely distributed pathogens that cause distinct clinical conditions in cattle including infectious bovine rhinotracheitis, infectious pustular vulvovaginitis/balanoposthitis, and meningoencephalitis. Due to the establishment of viral latency, controlling these infections is challenging, and vaccination remains the most effective strategy. In this study, vaccine candidates targeting both BoAHV-1 and BoAHV-5 were developed. Methods: A synthetic gene encoding immunodominant epitopes from the gB and gD proteins and tegument phosphoprotein of BoAHV-1 and BoAHV-5 was designed to produce a multi-epitope recombinant antigen, expressed both in a prokaryotic system (RecBoAHV) and by a modified vaccinia Ankara (MVA-BoAHV) viral vector. The binding affinity of MHC-I to bovine leukocyte antigens (BoLA) was predicted using the NetMHCpan tool (version 4.1). The immunogenicity of the vaccine candidates was evaluated in rabbit and mouse models, using prime-boost immunization protocols. Sera from bovines naturally infected with BoAHV-1 and/or BoAHV-5 were used to evaluate the chimeric protein antigenicity. Immune responses were assessed by indirect ELISA and Western blot. Results: The recombinant multi-epitope protein was effectively recognized by IgG and IgM antibodies in sera from cattle naturally infected with BoAHV-1 or BoAHV-5, confirming the antigenic specificity. Both RecBoAHV and MVA-RecBoAHV induced strong and specific humoral immune responses in rabbits following a homologous prime-boost regimen. In mice, both homologous and heterologous prime-boost protocols revealed robust immunogenicity, particularly after the second booster dose. Conclusions: These findings highlight the immunogenic potential of the RecBoAHV multi-epitope vaccine candidates for controlling BoAHV-1 and BoAHV-5 infections. Further characterization of these vaccine formulations is currently underway in bovine, the target specie.

## 1. Introduction

The bovine milk and beef industries, along with other derived products, play a vital role in the economies of many countries. Consequently, diseases that affect dairy or beef cattle are of major concern because of the significant economic impact they cause [1]. *Varicellovirus bovinealpha1* (previously *Bovine alphaherspervirus1*/infectious bovine rhinotracheitis virus; BoAHV-1) and *Varicellovirus bovinealpha5* (previously *Bovine alphaherspervirus 5*/bovine encephalitis herpesvirus; BoAHV-5) are two closely related members of the *Orthoherpesviridae* family, *Alphaherpesvirinae* subfamily [2,3]. Both viruses infect cattle and cause diseases in the respiratory and reproductive tracts as well as fetal infections and neonatal disease [4]. BoAHV-1 respiratory infections are commonly associated with infectious bovine rhinotracheitis (IBR), abortion, and systemic disease in neonates [5,6]. In contrast, BoAHV-5 infections primarily cause respiratory and neurological disorders including central nervous system diseases such as meningoencephalitis [6]. Both viruses are neurotropic and can establish a lifelong latent infection in neurons of sensory ganglia. Thus, herpesvirus infections are of great importance not only because of their acute clinical manifestations, which lead to direct economic losses, but also due to their ability to establish latency and reactivate from time to time, allowing viral perpetuation in nature and in commercial herds [6,7].

Most commercial vaccines against bovine herpesviruses are produced as multivalent formulas that contain several viral and bacterial antigens. Also, commercially available vaccines and vaccine candidates consist of attenuated or whole inactivated pathogens, or a single target protein [8]. The combination of subunit vaccines and viral vectors represents a promising approach for developing next-generation vaccines [9,10,11]. Subunit-based formulations and recombinant viral platforms, such as modified vaccinia Ankara (MVA), offer significant advantages by enabling the targeted inclusion of well-characterized immunogenic epitopes into a tailored antigen. These platforms are capable of inducing robust and balanced cellular and humoral immune responses while maintaining a favorable safety profile, as they are non-replicative in mammalian hosts and do not revert to virulence [12,13,14,15].

We have designed a recombinant multi-epitope, chimeric protein, consisting of four immunodominant, linear, and conserved epitopes from proteins of BoAHV-1 and BoAHV-5. Based on in silico analysis and data from the literature, we were able to define the most immunogenic epitopes among structural and non-structural viral proteins of the two viruses. The selected proteins were the glycoproteins gB and gD due to their key roles in infection, strong immunogenicity, and potential for vaccine development as well as the non-structural tegument phosphoprotein (UL47), which effectively induces cellular immune responses and elicits some humoral response. The genes encoding these epitopes were assembled in tandem in a single open reading frame, flanked by flexible linkers, and expressed in *Escherichia coli* as well as in a poxviral platform using a recombinant modified virus vaccinia Ankara, both capable of inducing humoral and cellular response (patent PI 1101186-6). The antigenicity of the purified protein was first evaluated in Western blot and indirect ELISA assays using the sera of naturally infected bovines. The immunogens were assessed in rabbit and mouse immunization models, where they elicited robust immune responses, underscoring the efficacy of the constructs.

## 2. Materials and Methods

### 2.1. Animals

New Zealand White male rabbits (*Oryctolagus cuniculus*) and C57BL/6 female mice were obtained from the animal facilities at Universidade Federal de Minas Gerais—Brazil (UFMG). Rabbits were housed in individual cages and mice in mini-isolators within ventilated racks at the maintenance facility. All animals were maintained under a controlled light/dark cycle with ad libitum access to food and water. The immunization procedures were conducted in accordance with the ethical principles of animal experimentation adopted by the Ethics Committee for Animal Experiments (CEUA/UFMG), as outlined in protocols CETEA #239/2011 and CEUA #350/2018. The CEUA/UFMG follows all Institutional Animal Care and Use Committee (IACUC) guidelines to minimize animal suffering. Group sizes were chosen after a review of the literature on bovine alphaherpesvirus preclinical studies and in-house pilot data. Several published BoAHV studies have used six mice/group per group for immunogenicity analyses [16,17,18] and small rabbit cohorts (two animals/group) for preliminary immunogenicity and protection tests [19,20]. These sample sizes balance statistical sensitivity with the 3Rs principle [21] of reduction and the stepwise approach from pilot testing to larger confirmatory trials.

### 2.2. In Silico Prediction of Immunodominant Epitopes and Construction of Synthetic Genes

The main bovine immunodominant B- or T-cell epitopes on glycoproteins gB, gD, and the non-structural tegument phosphoprotein of the BoAHV-1 (complete genome—GenBank: AJ004801.1) and BoAHV-5 (complete genome—GenBank: AY261359.1) were determined upon an extensive literature review and analyzed using the SYFPEITHI platform “https://syfpeithi.de/” (accessed on 10 March 2024) [22]. Four immunodominant epitopes, conserved in both viruses, were used to design a multi-epitope chimeric protein comprising one concomitant B- and T-cell epitope from the gB glycoprotein, two epitopes recognized by B and T cells from gD, and one T-cell epitope from the tegument phosphoprotein. Hydrophilic regions within the epitopes were identified using the DAS program [23] from the Expasy platform “https://www.expasy.org/” (accessed on 18 March 2024). The sequence was assembled in a tandem array with the addition of nucleotide sequences coding for flexible glycine-serine intergenic linkers between each epitope. The three-dimensional structure of the recombinant protein was predicted using the I-TASSER server “http://zhanglab.ccmb.med.umich.edu/I-TASSER” (accessed on 2 April 2024) [24]. The synthetic gene, encoding a recombinant BoAHV-1 and BoAHV-5 multi-epitope protein (RecBoAHV) was custom-synthesized by Entelechon (Baviera, Germany) and subcloned into the pGEMT-easy vector system (Promega—Madison, WI, USA). Primers for direct PCR amplification and insertion of BamHI and HindIII restriction enzyme sites to the 5′ and 3′ ends of the construct (5′ ATGGATCCCACCGCGAGCACACC 3′ and 5′ ATAAGCTTGTCGCTGCTATCGCCGTC 3′, respectively) were synthesized by Integrated DNA Technologies (San Diego, CA, USA).

### 2.3. In Silico Evaluation of the Immunogenic Coverage of the RecBoAHV via T- and B-Cell Epitope Prediction

The in silico prediction of T-cell and B-cell epitopes was performed to assess the immunogenic potential of the designed multi-epitope protein. Predictions of major histocompatibility complex (MHC) class I binding properties, specific for bovine leukocyte antigens (BoLA), were carried out using the NetMHCpan 4.1 tool (DTU Health Tech—Bioinformatic Services, “https://services.healthtech.dtu.dk/services/NetMHCpan-4.1/” accessed on 8 May 2025) [25]. The analysis included all available BoLA haplotypes present in the tool’s database, intending to evaluate whether the vaccine construct would provide broad immunological coverage across different bovine genetic backgrounds. Default parameters were applied in the in silico analyses, and the predicted peptides were classified into strong and weak binders according to the tool’s ranking thresholds. For the prediction of linear B-cell epitopes, the BepiPred-3.0 tool online source “https://services.healthtech.dtu.dk/services/BepiPred-3.0/” (accessed on 27 July 2025) [26] was employed. This method combines a machine learning approach with structural features to predict potential antibody-binding regions within the recombinant protein. The analysis was conducted using default thresholds and aimed to assess whether the final construct would elicit an effective B-cell-mediated humoral immune response.

### 2.4. Cloning of the Synthetic Gene RecBoAHV for Expression in a Prokaryotic Vector

The synthetic gene encoding the recombinant protein from the pGEMT-easy vector and the expression plasmid pQE-30 (Genscript—Piscataway, NJ, USA) were restricted with BamHI and HindIII enzymes (Promega—Madison, WI, USA). The DNA fragment of the RecBoAHV multi-epitope protein was purified and cloned into pQE-30. The pQE-30-RecBoAHV plasmid contains a T5 promoter for IPTG-inducible expression and an *N*-terminal 6 × His tag for affinity purification, and was used to transform the competent *E. coli* M15 strain (see Appendix A). Protein expression was induced with 1 mM isopropyl-β-D-1-thiogalactopyranoside (IPTG) for 5 h at 37 °C with shaking at 200 rpm. Following induction, bacterial cells were harvested, and the resulting pellet was lysed in Buffer A (30 mM imidazole, 8 M urea, 10 mM Tris-HCl, 100 mM NaH_2_PO_4_, pH 8.0) containing a protease inhibitor cocktail [5 mM dithiothreitol (DTT) and 1 mM phenylmethylsulfonyl fluoride (PMSF)]. The lysate was centrifuged at 6000× *g* for 1 h at 4 °C, and the supernatant was subjected to affinity chromatography using an EUR KTA Start system (GE Healthcare—Buckinghamshire, UK) equipped with a HisTrap column (Cytiva—Marlborough, MA, USA). Purified fractions were collected and analyzed by SDS–PAGE.

### 2.5. Construction of the Recombinant MVA-RecBoAHV Vector

The MVA used in these constructions is derived from the MVA-1974 clone, kindly provided by Dr. Bernard Moss (LVD/NIAID/NIH). This low-passaged MVA was transferred to UFMG under a specific material transfer agreement (MTA). The DNA fragment encoding the RecBoAHV multi-epitope protein initially contained in the pGEMT-easy vector was restricted with BamHI and HindIII enzymes (Promega—Madison, WI, USA). The fragment was inserted into the pLW44 transfer plasmid using T/A cloning and confirmed by sequencing (see Appendix A). The pLW44 transfer plasmid contains the green fluorescent protein (GFP) coding sequence under the control of the mH5 early/late *Orthopoxvirus vaccinia* (VACV) promoter [27]. BHK-21 cells (baby hamster kidney fibroblasts, obtained from the American Type Culture Collection—ATCC^®^ CCL-10™) were infected with MVA and subsequently transfected with pLW44-RecBoAHV using Lipofectamine 3000 Reagent (Invitrogen—Carlsbad, CA, USA). The construction of the recombinant vaccine vector was based on the homologous recombination between the plasmid (pLW44-RecBoAHV) and MVA. Cells infected with recombinant vector clones were selected with the aid of GFP expression, and clones expressing MVA-RecBoAHV were subsequently isolated. The MVA-RecBoAHV construct was validated by Sanger sequencing using the BigDye^®^ Terminator v1.1 Cycle Sequencing Kit (Applied Biosystems—Foster City, CA, USA) on a MegaBACE™ 1000 capillary sequencer (GE Healthcare—Buckinghamshire, UK) with primers annealing to the pLW44 transfer plasmid flanking regions (5′ AAAGACCCCAACGAGAAGC 3′ and 5′ GTCTGAGGAAAAGGTGTAGCG 3′). Sequence data were analyzed using Chromas v2.23 software “https://technelysium.com.au” accessed on 8 September 2024) and aligned with reference sequences in GenBank using BLASTn 2.17.0 version (NCBI, USA—”https://blast.ncbi.nlm.nih.gov/Blast.cgi” accessed on 6 February 2025). Transcript expression in infected BHK-21 cells was detected by RT-PCR using oligo(dT) and BoAHV-specific primers (5′ ATGTCGACCACCGCGAGCACACC 3′ and 5′ AGCTGCAGGTCGCTGCTATCGCC 3′). For the viral stock production, cells were infected at 37 °C, 5% CO_2_ atmosphere for 1 h. After adsorption, the cells were incubated in the same atmosphere in DMEM (Dulbecco’s modified Eagle medium—Sigma Aldrich—St. Louis, MO, USA) supplemented with 7.5% NaHCO_3_, antibiotics (100 μg/mL streptomycin and 100 U/mL penicillin), antifungal (fungizone at 25 μg/mL), and 5% fetal bovine serum (FBS-Gibco—Billings, MA, USA) for 48 h. Recombinant virus purification was performed in a 36% (*w*/*v*) sucrose cushion (in 10 mM Tris-HCl, pH 9.0), and the titer was determined by a plaque-based assay [28,29].

### 2.6. Immunization of New Zealand Rabbits

#### 2.6.1. Immunization with RecBoAHV

Four New Zealand White male rabbits, 7–8 weeks of age, weighing 2–2.5 kg, were kept in individual cages in a conventional animal facility with food and water ad libitum. The rabbits were divided into two groups (two animals/group) and immunized in a prime-boost-boost homologue protocol consisting of three subcutaneous doses at 21-day intervals. Group 1 received 1 mL of a solution composed of 5% aluminum hydroxide, 25% Emulsigen, and 70% recombinant protein (250 ng of RecBoAHV), whereas Group 2 was inoculated with 1 mL of phosphate-buffered solution as a negative control. Animals were bled immediately before vaccination (pre-vaccination) and after the prime-boost-boost dosing. Bleeding of the animals was performed by puncture in the marginal ear vein.

#### 2.6.2. Immunization with MVA-RecBoAHV in Prime-Boost-Boost Homologous Protocol, and Evaluation of Previous Anti-Poxvirus Immunity Interference

Twelve New Zealand White male rabbits, 7–8 weeks of age, weighing 2–2.5 kg, were kept in individual cages in a conventional animal facility with food and water ad libitum. For the evaluation of the MVA-RecBoAHV construct, animals were divided into three groups, with two animals per group. Initially, Group 1 received PBS, Group 2 received 1.0 × 10^7^ PFU MVA-GFP, and Group 3 received 1.0 × 10^7^ PFU of inactivated VACV strain Western Reserve (WR). After 14 days, all animals were administered 1.0 × 10^7^ PFU of MVA-RecBoAHV with the addition of PBS control (unvaccinated) and RecBoAHV-immunized groups (three animals/group). Rabbits were immunized following a prime-boost-boost protocol with a 14-day interval between each inoculation.

### 2.7. Immunization of Mice with MVA-RecBoAHV in Prime-Boost-Boost Protocols

C57BL/6 mice, 4 to 6 weeks old, were kept in microisolators with food and water ad libitum. Mice were divided into six groups (6 animals/group) and immunized intranasally according to a prime-boost-boost protocol consisting of three doses administered at 14-day intervals (Table 1). The animals were bled by puncturing the facial vein, and before euthanasia via the brachial plexus. The blood was tested in pools of each group. MVA immunizations were defined at a dose of 1 × 10^7^ PFU, in accordance with several previous studies in mice [30,31,32]. Protein immunization was defined as 1.0 µg/dose. The dose choice was based on the small size of the chimeric protein (~100 amino acids), providing an adequate molar ratio of protein molecules per dose. According to the UCSF Office of Research (University of California San Francisco), the dose recommended for the intranasal instillation of mice is 50 µL maximum volume, and following this recommendation, we prepared individual doses of 10 µL (divided equally between both nostrils).

### 2.8. Indirect ELISA

An indirect in-house ELISA [33] was performed to evaluate the recognition of the recombinant multi-epitope protein by IgG and IgM present in serum samples from a herd naturally infected and seropositive for BoHV-1 and BoHV-5 (previously analyzed by virus neutralization tests). Purified BoAHV-1 or BoAHV-5 (total viral proteins) (250 ng/well) or the recombinant protein RecBoAHV was diluted in 0.05 M carbonate-bicarbonate buffer (pH = 9.6) and used to coat polystyrene plates (Costar^®^ 9018, EIA/RIA plate, certified high binding; Corning, NY, USA) overnight at 4 °C. After adsorption of the antigens, plates were washed once with PBS-T (0.1% nonfat milk powder, 0.05% Tween-20 in phosphate buffered saline pH 7.2) and blocked with 150 µL/well of 5% nonfat milk powder in PBS overnight at 4 °C. The wells were further washed ten times with PBS-T and 100 µL of serum samples diluted 1:100 were added to each well. The plates were incubated at 37 °C for 1 h in a humid chamber, washed ten times with PBS-T, and 100 µL of anti-bovine IgG or IgM peroxidase conjugate (1:30,000 dilution, Sigma-Aldrich—St. Louis, MO, USA) was added/well. They were further incubated for 1 h at 37 °C in a humid chamber, washed ten times with PBS-T and 100 µL of OPD solution (o-phenylenediamine, Sigma-Aldrich—St. Louis, MO, USA) were added/well. The plates were incubated at 25 °C for 30 min in a dark chamber, and 30 µL of the stop solution (H_2_S0_4_) was added/well. The optical density (OD) was recorded at 492 nm in a microplate reader (Expert Plus Microplate Reader—G020 150, Asys Hitech-Eugendorf, Salzburgo, Austria). Tests were performed using duplicates of each serum. The same protocol was used to evaluate the reactivity of serum samples from the immunized rabbits against the recombinant protein and viral proteins of BoAHV-1, BoAHV-5, and MVA-RecBoAHV. The secondary antibody (anti-rabbit IgG-peroxidase) was diluted 1:20,000. The ELISA index was calculated as the ratio of sample OD to the cutoff, with the cutoff defined as the mean OD of negative controls plus two standard deviations. Samples with index values > 1.1 were considered positive, <0.8 negative, and those between 0.8 and 1.1 indeterminate.

### 2.9. Western Blot

To detect recombinant protein and BoAHV-1/BoAHV-5 viral proteins using antibodies from the sera of immunized rabbits and mice or from naturally infected cattle, 10 µg of each protein was separated on a 12% SDS-PAGE gel alongside pre-stained protein markers and then transferred to polyvinylidene difluoride membranes. The membranes were blocked with 5% nonfat milk powder in PBS overnight at 4 °C. They were then washed three times with 1× PBST (phosphate-buffered saline [PBS], 0.1% Tween 20, pH 7.2) and incubated with diluted serum samples (1:20 dilution) overnight at 4 °C. The membranes were washed as described above and incubated with each respective secondary antibody [anti-bovine IgG peroxidase conjugate, anti-rabbit, or anti-mouse IgG-peroxidase (1:500 dilution)] for 2 h at 37 °C. After the wash, the protein bands were revealed with HRP Color Development Reagent (Bio-Rad—Hercules, CA, USA).

### 2.10. Statistical Analysis

Results were statistically evaluated using ANOVA with post-test Tukey and Kruskal–Wallis with Dunn post-tests, performed in GraphPad PRISM version 8.0.1, and *p* values < 0.05 among groups were considered significant.

## 3. Results

### 3.1. Design of the Recombinant Multi-Epitope Protein and Expression

B- and T-cell epitopes were identified using the SYFPEITHI epitope prediction platform and corroborated by data from the literature. Conserved linear epitopes from the gB, gD, and tegument phosphoprotein proteins of both BoAHV-1 and BoAHV-5 were selected and incorporated into the design of the recombinant multi-epitope protein (Table 2). These proteins were chosen not only because they are conserved among BoAHV-1 and -5, but also for their well-documented immunological relevance in BoAHV infection. Glycoprotein B (gB) and glycoprotein D (gD) are essential for viral adsorption, entry, and cell-to-cell spread, and have been shown to elicit robust humoral and cellular immune responses including the induction of neutralizing antibodies and CD4+/CD8+ T-cell activation. The tegument phosphoprotein, although less accessible to antibody recognition, is abundant in virions and capable of inducing strong T-cell responses, contributing to effective cellular immunity [23,24].

Hydrophobicity profiling and transmembrane domain prediction (see Appendix A) were performed to exclude highly hydrophobic sequences or regions potentially involved in membrane anchoring, which could impair epitope accessibility and immunogenicity. Due to the absence of a single region within gD containing both predicted T- and B-cell epitopes, two distinct segments were selected to ensure broader immune coverage. Regarding the tegument phosphoprotein, only T-cell epitopes were included in the construct, based on its exclusive intracellular localization during virus replication, which limits accessibility for extracellular antibody recognition.

Linear B-cell epitope prediction was performed using BepiPred 3.0. The analysis was applied to the designed RecBoAHV antigen, and the default threshold of 0.15 was used to identify residues with potential B-cell antigenicity. Remarkably, the entire vaccine sequence exceeded the positivity threshold, indicating a continuous profile of predicted B-cell epitope potential. Although all regions were classified as antigenic, the segment spanning amino acid residues 20 to 45, located within Pep1, presented the highest BepiPred scores. This suggests that this region may be particularly accessible and reactive to B-cell receptors, potentially serving as a major target for antibody recognition. Score variation along the sequence is illustrated in Figure 1A, highlighting this and other moderately reactive regions.

Prediction of the MHC-I binding epitopes was conducted in silico using the NetMHCpan 4.1 tool, targeting the most frequent BoLA class I haplotypes identified in cattle populations. The analysis identified multiple predicted epitopes with either strong or weak binding affinity across the vaccine antigen sequence (see Appendix A). According to standard classification thresholds, strong binders (SB) were defined as peptides with predicted affinity scores ≤ 0.5.

A total of 32 peptides were evaluated for binding across 62 BoLA haplotypes, distributed as follows: 15 in Pep1, 8 in Pep2, 6 in Pep3, and 3 in Pep4. As shown in Figure 1B, most predicted epitopes exhibited strong binding to multiple BoLA alleles, indicating the potential for broad immunogenetic coverage. Among them, the peptide YSPERFQQI showed strong binding to 32 distinct BoLA haplotypes, characterizing it as a promiscuous cytotoxic T lymphocyte (CTL) epitope. Additionally, the peptide SEDENVYDY, derived from Pep4, exhibited strong binding with the BoLA-1 and BoLA-6 haplotypes, suggesting potential for targeted antigen presentation in animals expressing these alleles.

When stratifying the BoLA-I haplotypes by family, a clear predominance of strong binder interactions was observed among the BoLA-3 alleles, which accounted for 24 out of the 62 haplotypes analyzed (approximately 39%). This group exhibited the highest frequency of strong binding peptides, indicating a potential predominance of antigen presentation through BoLA-3 molecules. This trend suggests that cattle expressing BoLA-3 alleles may exhibit enhanced responsiveness to the vaccine candidate, reinforcing the relevance of immunogenetic profiling in vaccine design.

The three-dimensional structure of the recombinant vaccine antigen was predicted using the I-TASSER 5.1 version software. The resulting structural model was visualized in three orientations, highlighting the spatial distribution of secondary structural elements including α-helices, β-strands, and coils (Figure 1C). The visualization revealed a protein architecture composed of defined helical and strand regions interspersed with flexible loops, suggesting a partially ordered structure compatible with proper folding.

### 3.2. Expression and Characterization of the Recombinant Multi-Epitope Protein

The RecBoAHV coding gene was subcloned into the pQE30 plasmid and used to transform *E. coli* M15 cells. The transformant bacteria were cultured and induced with IPTG to drive expression of the recombinant protein, and the expression was evaluated by SDS-PAGE (Figure 2A). As predicted, a recombinant multi-epitope protein of approximately 12.7 kDa was detected. A small amount of the protein was also observed in the uninduced fraction due to basal (“leaky”) expression, which is common in bacterial expression systems. The protein was efficiently purified using affinity chromatography on a Ni-NTA chelating column (Figure 2A), and fractions E1–E5 displayed a characteristic pattern of two bands, likely resulting from minor degradation, the presence of the initiating methionine, or different conformational states of the recombinant protein.

Western blot analysis (Figure 2B) demonstrated that a pool of bovine sera from animals naturally infected by BoAHV-1 and BoAHV-5 strongly recognized the recombinant multi-epitope protein, demonstrating the antigenicity of the generated protein and confirming that the selected epitopes are immunologically relevant in naturally infected animals. Furthermore, individual analysis by indirect ELISA for IgG and IgM confirmed that all bovine sera from BoHV-1/BoHV-5-positive animals were able to specifically recognize the recombinant protein, reinforcing the preservation of antigenic determinants and the potential for immunization purposes (Figure 2C). As expected, IgG reactivity was lower with the recombinant protein compared with the total antigen, since the recombinant protein contained only selected epitopes and thus recognized a narrower subset of antibodies.

### 3.3. Immunogenicity of the RecBoAHV in a Rabbit Model

Rabbits were immunized with the recombinant protein following a prime-boost-boost homologous protocol consisting of three subcutaneous doses given at 21-day intervals. Sera from these animals were collected 14 days after final immunization and evaluated by indirect IgG ELISA, showing reactivity to the recombinant protein as well as to the purified BoAHV-1 and BoAHV-5 (total viral proteins) immobilized on the solid phase (Figure 3). After the prime dose, a statistically significant difference was already observed between the immunized and control groups, with a marked increase following the second boost, after which high IgG and IgM antibody titers were detected, with OD values reaching above 2.0 for all three tested antigens. Sera from the control rabbits (that received PBS only) did not recognize any of the tested antigens.

### 3.4. Immunogenicity of the MVA-RecBoAHV in a Rabbit Model

For the immunization of rabbits with MVA-RecBoAHV, recombinant viruses were generated after homologous recombination between the MVA genome and the pLW44 transfer plasmid containing the RecBoAHV coding gene. Recurrent nucleotide sequencing of plasmids and viral DNA was carried throughout the recombinant viral vector construction process, confirming the correct insertion of the target gene into the MVA genome (see Appendix A).

The effect of pre-existing poxvirus immunity on the immunogenicity of MVA-RecBoAHV was evaluated using an indirect ELISA. Sera collected from rabbits 14 days after prime immunization with PBS, MVA-GFP, or VACV-WR were analyzed for IgG reactivity to BoAHV-1 and BoAHV-5. Next, the animals received the MVA-RecBoAHV immunogen in a homologous prime-boost-boost vaccination protocol. Serum samples from the animals were also compared with those from rabbits immunized only with the RecBoAHV protein or with PBS as controls. After receiving the second boost with MVA-RecBoAHV, all groups exhibited high levels of specific antibodies against BoAHV-1 and BoAHV-5, regardless of pre-existing immunity to vaccinia virus (MVA or VACV-WR) when compared with the group inoculated with PBS only (Figure 4). The group vaccinated with the recombinant protein also showed high levels of IgG against BoAHV-1 and BoAHV-5, as expected.

### 3.5. Immunogenicity of the RecBoAHV in a Mouse Model

Female C57BL/6 mice were used to evaluate the immunogenicity of the vaccine prototype. The animals were immunized via the intranasal route following a prime-boost-boost protocol (Table 1). The levels of IgM and IgG were measured by indirect ELISA (Figure 5).

Although mice are not the ideal model for evaluating vaccines intended for cattle, they represent a practical and accessible tool for complementary analyses, allowing a detailed characterization of the immunogenicity and immune response mechanisms induced by the vaccine candidates. To better understand these responses, a murine MHC haplotype binding prediction was performed (see Appendix A), demonstrating that, despite the limitations of this model, it remains relevant by showing a cellular response against the study’s target protein.

Vaccinated groups exhibited a trend toward the increased production of antigen-specific IgG antibodies compared with the PBS control and MVA-GFP groups during both the prime and first boost. The heterologous regimen (MVA-RecBoAHV) induced the highest IgG levels, surpassing not only the controls, but also the other vaccinated groups. A similar pattern was observed for IgM antibodies. Moreover, antibody titers against BoAHV-5 tended to be higher than those against BoAHV-1. Following the second boost, the heterologous protocol continued to elicit the highest titers, which were statistically significantly greater than those of all the other vaccinated groups. Although the MVA-based protocol elicited the strongest response, the recombinant protein also induced statistically significant increases compared with the PBS and MVA-GFP controls for BoAHV1 IgG as well as for BoAHV1 and BoAHV5 IgM. Notably, the group immunized with inactivated antigens displayed antibody levels statistically comparable to those induced by the recombinant protein.

Comparative analysis of ELISA indices for IgG and IgM revealed distinct responses across the animal models and immunization regimens (Table 3). In rabbits immunized with RecBoAHV, robust IgG responses were observed against BoAHV-1 and BoAHV-5, with indices reaching 7.89 and 6.22, respectively, after the second boost. In contrast, rabbits vaccinated with MVA-RecBoAHV HO (homologous protocol—prime and boosts with MVA-RecBoAHV) exhibited more moderate responses, with indices of 2.94 against BoAHV-1 and 5.39 against BoAHV-5 following the second boost.

In the mice experiments, all immunizing agents induced seroconversion for both IgG and IgM, with indices above the cutoff as early as the priming dose. Animals immunized with RecBoAHV exhibited a progressive increase in the IgG and IgM indices against both viruses, reaching up to 1.75 (IgG anti-BoAHV-5) and 1.72 (IgM anti-BoAHV-1) after the second boost. Vaccination with MVA-RecBoAHV HO induced moderate responses, with IgG indices ranging from 1.11 to 1.85 and IgM indices from 1.19 to 1.57. MVA-RecBoAHV HE (heterologous protocol—prime with RecBoAHV and boosts with MVA-RecBoAHV) elicited the highest serological responses, particularly for IgG anti-BoAHV-5 (3.90) and IgM anti-BoAHV-5 (4.80) after the first boost. The inactivated BoAHV-1/5 vaccine promoted intermediate levels of IgG (up to 2.00) and IgM (up to 2.16) against both viruses.

## 4. Discussion

BoAHV-1 and BoAHV-5 are pathogens of relevance to bovine health, being associated with respiratory, reproductive, and neurological diseases that result in significant economic losses in livestock production worldwide [34,35]. Although commercial vaccines are available, many of these formulations consist of combinations of viral and bacterial antigens, and efficacy levels vary [8]. Although the use of multivalent vaccines represents a practical strategy for herd management, enabling multiple immunizations through a single application, the simultaneous administration of multiple antigens may compromise the specific immune response to individual components. This phenomenon can be attributed to immunodominance, in which more immunogenic antigens elicit stronger immune responses, thereby suppressing or reducing the response to less immunogenic ones [36].

The development of multi-epitope vaccines and recombinant platforms is a promising strategy that has been increasingly evaluated [37,38,39,40,41]. It allows the immune response to be directed toward conserved regions of the viruses capable of stimulating both humoral and cellular responses [42]. The careful selection of conserved epitopes from the gB, gD, and tegument phosphoprotein proteins enabled the design of a recombinant antigen with broad immunogenic potential. The identification of epitopes with high affinity for multiple BoLA haplotypes is essential for the development of vaccines applicable to genetically diverse herds [43,44]. The peptide YSPERFQQI (from peptide 1—gB protein—B and T cells induction; Table 2), for example, showed strong affinity to more than 30 BoLA alleles, characterizing it as a promiscuous epitope with high potential to induce a robust cellular response in different genetic backgrounds. This broad reactivity suggests that the vaccine formulation has the potential to overcome the limitations imposed by MHC diversity, one of the main challenges in immunizing heterogeneous populations. The predominance of strong-binding epitopes among the BoLA-3 haplotypes may reflect structural or sequence-specific characteristics of these alleles that favor epitope presentation [45]. However, this reactivity was not restricted to a single group, as epitopes were also recognized by several other BoLA haplotypes, reinforcing the immunogenetic coverage of the proposed formulation. Likewise, B-cell epitope prediction revealed a consistently antigenic profile in the vaccine protein. Taken together, these analyses highlight the vaccine potential for inducing broad immune responses.

These observations are consistent with the general genomic stability observed in BoAHV-1 and BoAHV-5. Studies of whole-genome sequencing have shown that essential structural genes, such as gB, gC, gD, gH, and gM, are highly conserved across viral strains, whereas most genetic variation tends to accumulate in non-essential or surface-exposed regions of glycoproteins, potentially contributing to some antigenic diversity [46,47,48]. This overall pattern is consistent with the alignment results presented here, in which epitopes derived from gB (Pep1) and tegument phosphoprotein (Pep4) were fully conserved, while gD-derived epitopes (Pep2 and Pep3) showed strain- or genotype-associated divergence, particularly in the BoAHV-5 sequences. The inclusion of these conserved epitopes in vaccine design is therefore expected to enhance cross-protective immunity (see Appendix A).

Recognition of the purified protein by sera from naturally infected cattle confirmed the preservation of its antigenic specificity. Immunological assays in rabbits demonstrated that the prime-boost-boost immunization protocol elicited a robust humoral response, with elevated levels of IgG and IgM antibodies against BoAHV-1 and BoAHV-5 antigens, as detected by ELISA, for both the RecBoAHV and MVA-RecBoAHV formulations. The immunization data obtained with MVA-RecBoAHV further confirm that the RecBoAHV recombinant protein was effectively expressed by the viral vector, inducing a significant and specific immune response in immunized animals. Altogether, these preliminary findings support the immunogen’s potential as a vaccine antigen candidate, targeting both BoAHV-1 and BoAHV-5. Nevertheless, clinical experiments in cattle, the target species, are crucial to confirm the vaccine efficacy potential.

The MVA vector is known to be an excellent inducer of anamnestic responses [49], acting more effectively when boosting a response initiated by another homologous immunogen (in this case, the purified recombinant protein produced in prokaryotes). Because MVA is unable to complete its replication cycle in host cells [50], MVA induces only short-term antigen production, making it more suitable as a boosting element.

In a few countries, such as Brazil and India, for instance, feral strains of vaccinia virus are known to circulate and cause infections in cattle [51,52,53]. Thus, it was important to know whether pre-existing immunity to poxvirus (especially vaccinia viruses) could affect the immunogenicity of MVA-RecBoAHV. Pre-existing immunity against viral vectors can lead to early vector neutralization, compromising antigen expression and thus reducing vaccine efficacy. Our results showed that rabbits vaccinated with MVA-RecBoAHV exhibited high levels of specific antibodies against BoAHV-1 and BoAHV-5, regardless of the previous administration of MVA-GFP or VACV-WR.

Similarly to the rabbit vaccination model, mice intranasally immunized with the recombinant MVA-RecBoAHV vector produced robust and consistent immune responses, with the heterologous protocol standing out by inducing significantly higher levels of IgG and IgM antibodies compared with all other experimental groups, suggesting both strong humoral activation and potential functional relevance for protection. It is important to note, however, that the recombinant protein also induced statistically significant responses. Although these responses were lower than those elicited by the heterologous protocol, this finding is noteworthy, as simpler alternatives in terms of production, storage, and transport of the vaccine may be required for its practical use.

As mentioned previously, mice are not the optimal model for vaccine studies targeting cattle. When comparing T-cell epitope predictions, all four peptides exhibited numerous strong binders for BoLA (Pep1 > Pep2 > Pep4 > Pep3). However, only Pep1 (gB) and Pep4 (tegument phosphoprotein) showed strong binders in the mouse genetic background, Pep2 displayed only weak binders, and Pep3 showed none. These results suggest that the immune response in the target species is likely to be more robust than that observed in the murine model.

Heterologous vaccination strategies have been widely explored for their ability to promote more robust and long-lasting immune responses by combining distinct immunological stimuli. By employing different vaccine platforms for the priming and boosting doses, it is possible to enhance both humoral and cellular immunity while avoiding anti-vector immunity that may arise from the repeated use of the same vector platform in homologous regimens. Recent studies on COVID-19 vaccines have shown that heterologous combinations, such as Oxford/AstraZeneca followed by Pfizer/BioNTech, elicit stronger and longer-lasting immune responses compared with homologous regimens [54]. This approach has also proven promising in preclinical models for diseases, reinforcing its applicability across various immunological contexts and species [13,55]. Thus, the results obtained in this study further support the efficacy of the heterologous strategy in eliciting a significant immune response against both BoAHV-1 and BoAHV-5.

A tuberculosis booster study highlighted the importance of the intranasal delivery of recombinant MVA in eliciting protective immunity [56]. Like many pathogens, bovine herpesviruses infect via mucosal surfaces, underscoring the need for vaccine strategies that induce strong mucosal immune responses. Additionally, Belyakov et al. (1999) [57] demonstrated that mucosal immunization can overcome pre-existing immunity to poxviruses. Other studies have also reported promising results using an MVA-based vaccine expressing a secreted form of the BoAHV-1 gD protein, with immunization performed via the intranasal route [17,58].

Although specific cellular responses remain to be fully characterized, our data suggest that the combination of recombinant multi-epitope protein and a viral vector is a promising strategy for inducing broad and potentially effective immunity. The inclusion of appropriate adjuvants may enhance immunogenicity and broaden the protective potential of the formulation, representing a promising path for practical applications in cattle immunization. Future investigations should address these gaps including challenge studies with the virus in relevant animal models. Although protection was not directly assessed via viral challenge in the present study, the observed immunogenic profile supports the rationale for conducting further efficacy trials in cattle. Indeed, a study of this vaccine’s immunogenicity and efficacy in bovines is currently being conducted. A further limitation of the present study is the lack of virus neutralization assays, which are essential to evaluate the functional capacity of vaccine-induced antibodies. To address this, virus neutralization assays are currently being performed in cattle, and the results will provide critical complementary data to the immunogenicity findings reported here.

## 5. Conclusions

This vaccine candidate, based on conserved epitopes from BoAHV-1.1, BoAHV-1.2, and BoAHV-5 structural (gB, gD) and non-structural tegument phosphoprotein, enabled the design of a multi-epitope recombinant antigen with broad immunogenic potential. Predicted B-cell epitopes with high affinity for multiple BoLA haplotypes were confirmed by recognition of the recombinant protein by sera from naturally infected cattle. In pre-clinical studies, prime-boost homologous and heterologous vaccination with RecBoAHV and MVA-RecBoAHV elicited robust humoral responses, including elevated IgG in rabbits and IgG/IgM in mice against BoAHV-1 and BoAHV-5, highlighting the vaccine’s potential. Further evaluation of cellular immunity, IgG subclasses, neutralizing antibodies, and cytokine responses is underway in ongoing bovine studies.

## 6. Patents

This work has resulted in the granting of patent PI11011866.

## Figures and Tables

**Figure 1 vaccines-13-01115-f001:**
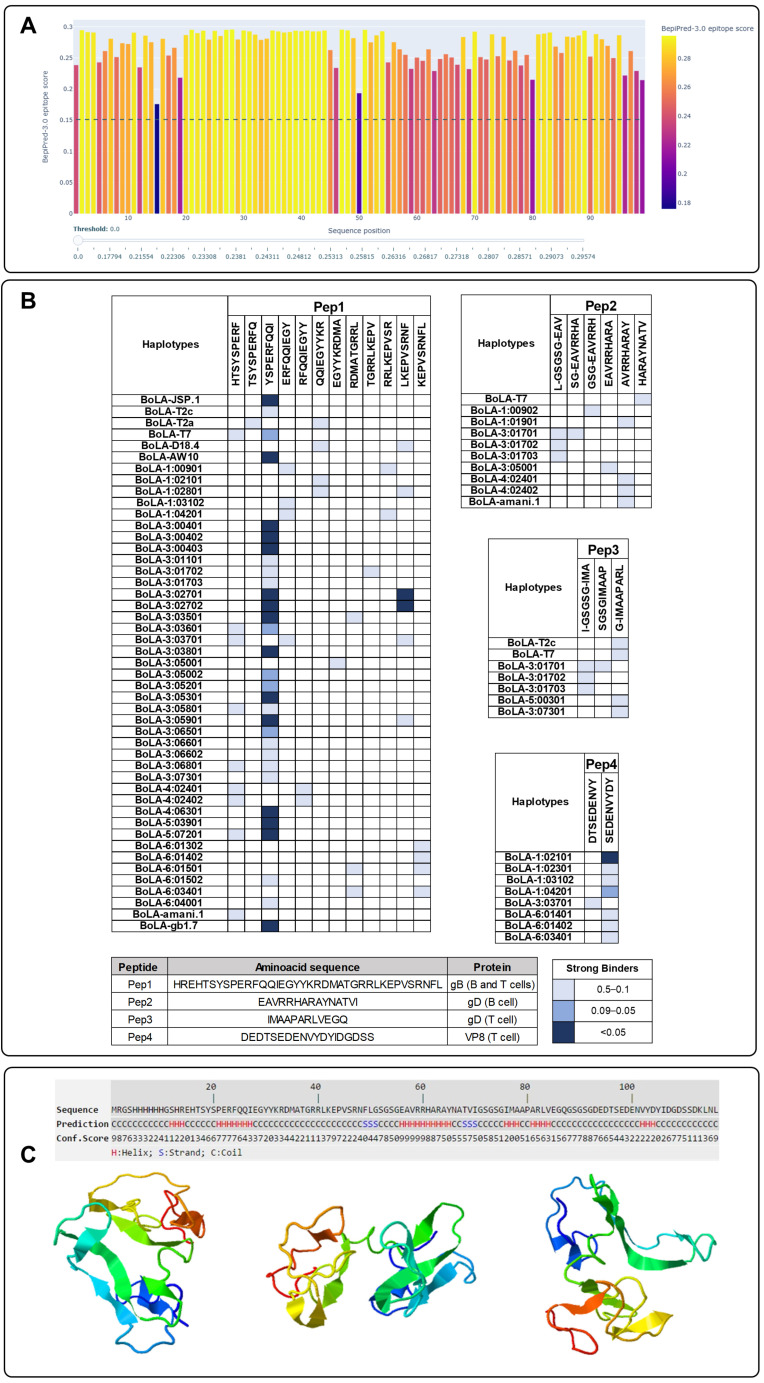
In silico evaluation of the RecBoAHV multi-epitope construct. (**A**) Prediction of linear B-cell epitopes using the BepiPred 3.0 tool. The full amino acid sequence of the RecBoAHV construct was analyzed, with residues scoring above the default threshold of 0.15 considered as potential antigenic sites. Regions exceeding the threshold are shown, with a prominent peak corresponding to residues 20–45 (within Pep1), which displayed the highest B-cell epitope scores. (**B**) Prediction of T-cell epitopes based on binding affinity to bovine leukocyte antigen (BoLA) class I haplotypes, using the NetMHCpan 4.1 tool. A total of 32 peptides was analyzed across 62 BoLA haplotypes (15 in Pep1, 8 in Pep2, 6 in Pep3, and 3 in Pep4). Strong binders (SB) were defined as peptides with predicted affinity ≤ 0.5 and are indicated in the heatmap. (**C**) Predicted three-dimensional structure of the RecBoAHV protein, generated by I-TASSER (I-TASSER server for protein structure and function prediction). The model is shown in three orientations, with α-helices, β-strands, and loop regions annotated. Selected epitopes are mapped on the protein surface to illustrate their accessibility and structural context.

**Figure 2 vaccines-13-01115-f002:**
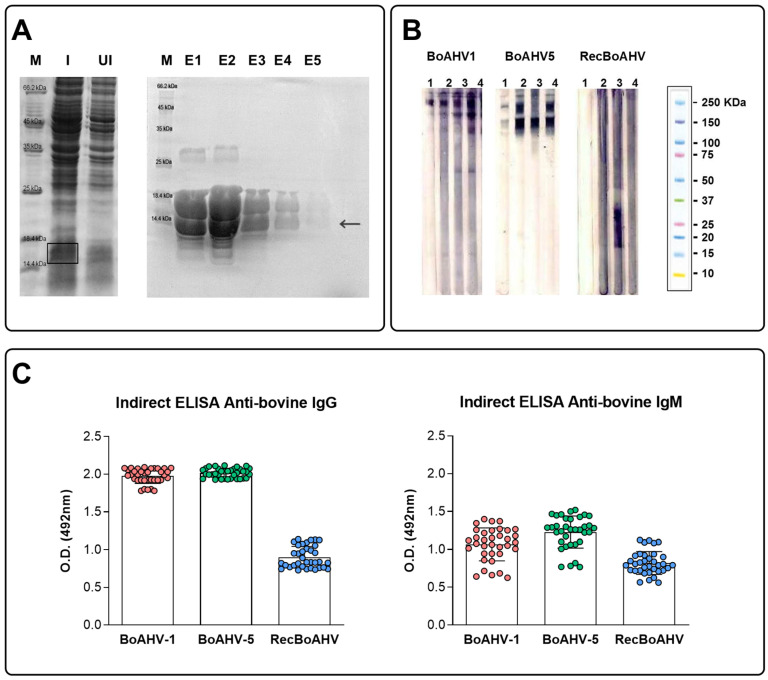
Characterization of RecBoAHV and reactivity with sera from naturally infected cattle. (**A**) Expression and purification of the RecBoAHV protein. The synthetic multi-epitope gene was cloned into the pQE-30 vector and expressed in *E. coli* M15 cells following IPTG induction. Protein extracts were analyzed by SDS–PAGE: (M) molecular weight marker (Sigma-Aldrich, USA); (UI) uninduced colony extract; (I) induced colony extract; (E1–E5) sequential fractions obtained after purification by Ni-NTA affinity chromatography. A distinct band corresponding to the expected size of ~12.7 kDa was observed in induced and purified fractions. The arrow and box represent the location of RecBoAHV on SDS-PAGE (**B**) Antigenicity of RecBoAHV assessed by Western blotting using bovine sera. Membranes were probed with: (1) serum from a BoAHV-1/BoAHV-5-negative animal; (2) serum highly positive for BoAHV-1; (3) serum highly positive for BoAHV-5; and (4) serum highly positive for both BoAHV-1 and BoAHV-5. The recombinant protein was specifically recognized by positive sera, confirming the immunological relevance of the selected epitopes. (**C**) Serological recognition of RecBoAHV by naturally infected cattle. IgG and IgM indirect ELISA were performed using bovine sera positive for BoAHV-1 and/or BoAHV-5. Positive sera consistently reacted with the recombinant protein for both immunoglobulins, reinforcing the preservation of its antigenic determinants and supporting its potential use for immunization or serological applications.

**Figure 3 vaccines-13-01115-f003:**
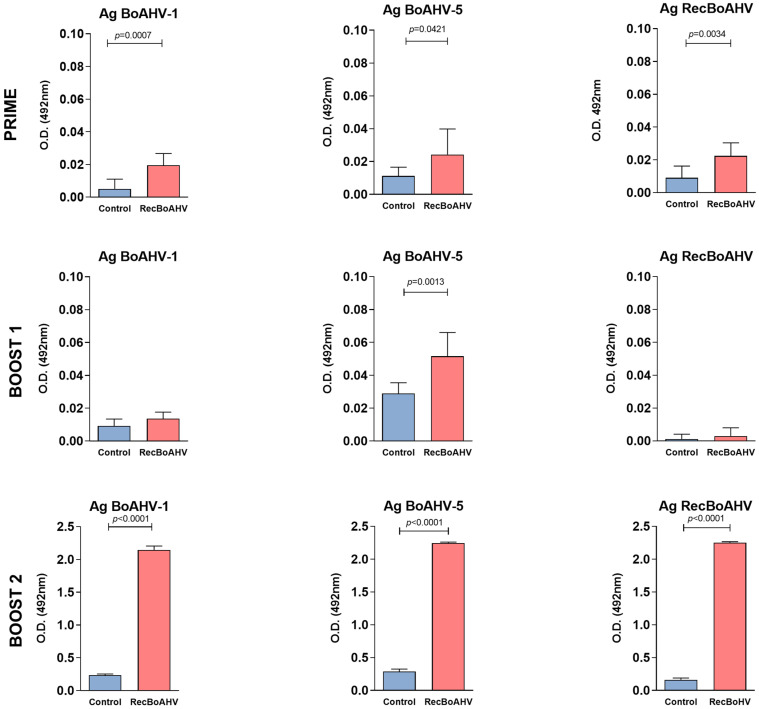
Evaluation of the antibody responses in New Zealand rabbits after vaccination with RecBoAHV. Indirect ELISA for anti-rabbit IgG detection using as antigens the recombinant multi-epitope protein (RecBoAHV), BoAHV-1, and BoAHV-5 purified viral proteins. Rabbits were immunized with RecBoAHV following a homologous prime-boost-boost protocol (three subcutaneous doses at 21-day intervals). Serum samples were collected at different time points (prime, boost 1, and boost 2) and analyzed for IgG reactivity. Statistically significant differences were observed between the immunized and control groups already after the prime dose, with a marked increase in antibody titers following the second boost. By boost 2, high IgG levels were detected against all tested antigens, with absorbance values above 2.0. Rabbits from the control group (PBS) showed no reactivity to any of the antigens. Statistical analysis was performed using the Kruskal–Wallis followed by Dunn’s multiple comparisons test. Each group contained two animals, which were tested in three replicates.

**Figure 4 vaccines-13-01115-f004:**
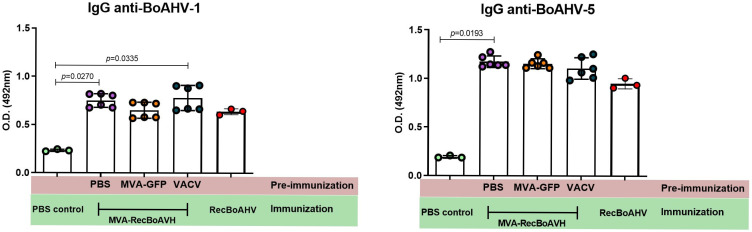
Evaluation of antibody responses in New Zealand rabbits after vaccination with MVA-RecBoAHV. Rabbits were pre-immunized with PBS, MVA-GFP, or VACV-WR to assess the potential effect of pre-existing poxvirus immunity on the immunogenicity of MVA-RecBoAHV. Subsequently, animals were immunized with MVA-RecBoAHV using a homologous prime-boost-boost vaccination scheme. Serum samples were collected 14 days after the second boost and analyzed by indirect ELISA for IgG reactivity against BoAHV-1 and BoAHV-5 purified viral proteins. As controls, groups of rabbits immunized only with RecBoAHV protein or PBS were included. Regardless of prior exposure to MVA or VACV-WR, animals vaccinated with MVA-RecBoAHV developed high levels of specific antibodies against both BoAHV-1 and BoAHV-5, comparable to the responses observed in the group immunized with RecBoAHV protein. In contrast, no significant reactivity was detected in the PBS control group. Statistical comparisons were performed using the Kruskal–Wallis followed by Dunn’s multiple comparisons test. The PBS, MVA-GFP, and VACV-WR groups were evaluated in two animals/group and tested in three replicates. The PBS control and RecBoAHV groups were evaluated in three animals and tested in single replicates.

**Figure 5 vaccines-13-01115-f005:**
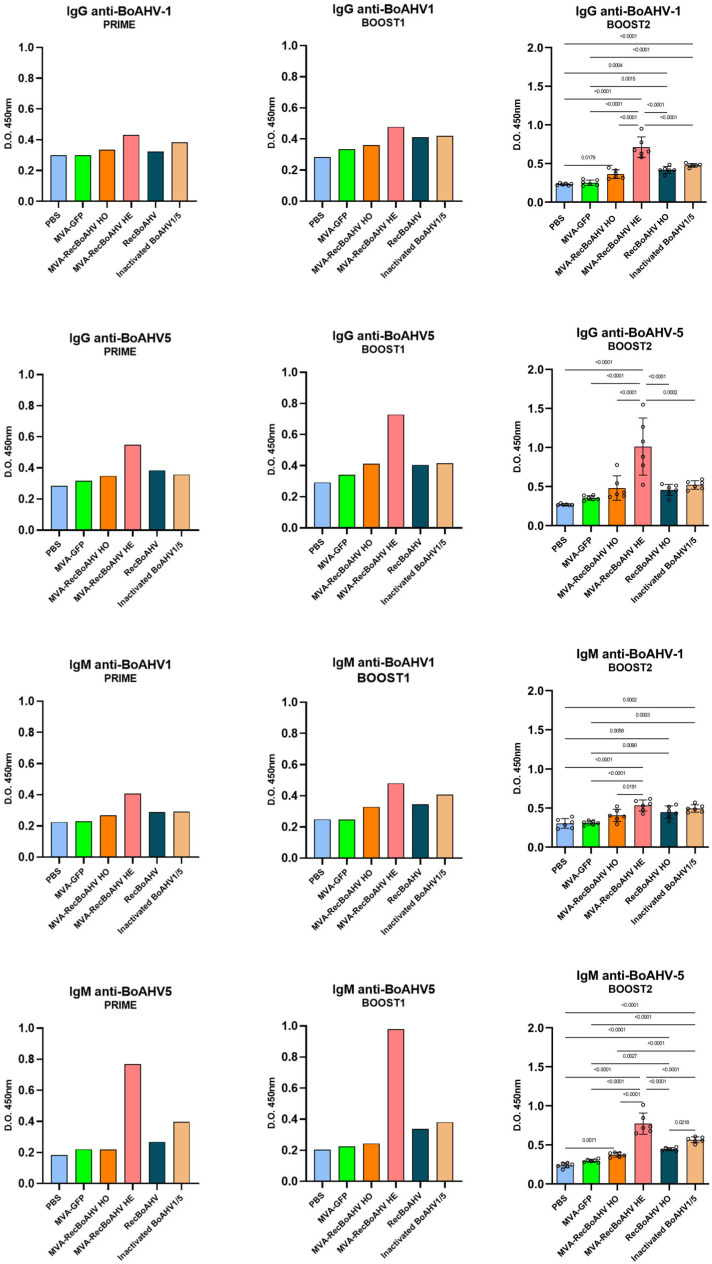
Evaluation of the antibody responses in C57BL/6 mice after vaccination. Indirect ELISA anti-bovine IgG and IgM for BoAHV1 and BoAHV5 antigens from mice immunized with MVA-RecBoAHV. The animals were immunized in a prime-boost-boost protocol. All groups consisted of six animals per group. Prime and Boost 1 were evaluated in a pool format, being tested in two technical replicates. Boost 2 was evaluated individually in single replicates. The statistics presented for the boost 2 results correspond as follows: IgG BoAHV1 was performed using ANOVA multiple comparisons test with post hoc Tukey’s, IgG BoAHV5 was performed using Kruskal–Wallis multiple comparisons test with post hoc Dunn’s, and IgM BoAHV5 was performed using ANOVA multiple comparisons test with the post hoc Tukey’s test.

**Table 1 vaccines-13-01115-t001:** Vaccination schedule of C57BL/6 mice.

Group	Prime	Boost 1	Boost 2
1	PBS	PBS	PBS
2	MVA-GFP 1.0 × 10^7^ PFU	MVA-GFP 1.0 × 10^7^ PFU	MVA-GFP 1.0 × 10^7^ PFU
3	MVA-RecBoAHV 1.0 × 10^7^ PFU	MVA-RecBoAHV 1.0 × 10^7^ PFU	MVA-RecBoAHV 1.0 × 10^7^ PFU
4	RecBoAHV 1.0 µg	MVA-RecBoAHV 1.0 × 10^7^ PFU	MVA-RecBoAHV 1.0 × 10^7^ PFU
5	RecBoAHV 1.0 µg	RecBoAHV 1.0 µg	RecBoAHV 1.0 µg
6	BoAHV-1 + BoAHV-5 (inactivated antigens) 1 × 10^6^ TCID_50_	BoAHV-1 + BoAHV-5 (inactivated antigens) 1 × 10^6^ TCID_50_	BoAHV-1 + BoAHV-5 (inactivated antigens) 1 × 10^6^ TCID_50_

**Table 2 vaccines-13-01115-t002:** BoAHV-1 (GenBank AJ004801.1) and BoAHV-5 (GenBank AY261359.1) specific immunodominant epitopes selected for the design of the multi-epitope recombinant protein.

Protein	Virus	Peptide	Amino Acid Sequence	Position (Protein ID)
gB (B and T cells)	BoAHV-1	Pep1	HREHTSYSPERFQQIEGYYKRDMATGRRLKEPVSRNFL	319–356 (CAA06106.1)
	BoAHV-5			326–363 (AAR86134.1)
gD (B cells)	BoAHV-1	Pep2	EAVRRHARAYNATVI	92–106 (CAA06145.1)
	BoAHV-5		IADPQVGRTLWGAVRRNARTYNATVIWYKIESGCA	82–116 (AAR86173.1)
gD (T cells)	BoAHV-1	Pep3	IMAAPARLVEGQ	161–172 (CAA06145.1)
	BoAHV-5		FAYPTDDELGLVMAAPARLAEGQYRRALYIDG	151–182 (AAR86173.1)
Tegument phosphoprotein (T cells)	BoAHV-1	Pep4	DEDTSEDENVYDYIDGDSSD	62–81 (CAA06087.1)
	BoAHV-5			62–81 (AAR86115.1)
RecBoAHV		HREHTSYSPERFQQIEGYYKRDMATGRRLKEPVSRNFLGSGSGEAVRRHARAYNATVIGSGSGIMAAPARLVEGQGSGSGDEDTSEDENVYDYIDGDSSD		

The underlined sequences refer to the linkers adding the final RecBoAHV sequence.

**Table 3 vaccines-13-01115-t003:** Comparative IgG and IgM ELISA indices between the animal models and immunizing conditions.

Animal Model	Immunizing Agent	ELISA IgG	Prime	Boost 1	Boost 2	ELISA IgM	Prime	Boost 1	Boost 2
Rabbit	RecBoAHV	Anti-BoAHV-1	1.14	0.77	7.89	-	-	-	-
		Anti-BoAHV-5	1.11	1.22	6.22	-	-	-	-
	MVA HO	Anti-BoAHV-1	-	-	2.94	-	-	-	-
		Anti-BoAHV-5	-	-	5.39	-	-	-	-
Mice	RecBoAHV	Anti-BoAHV-1	1.08	1.45	1.60	Anti-BoAHV-1	1.28	1.38	1.72
		Anti-BoAHV-5	1.35	1.38	1.75	Anti-BoAHV-5	1.45	1.65	1.71
	MVA-RecBoAHV HO	Anti-BoAHV-1	1.11	1.27	1.39	Anti-BoAHV-1	1.19	1.32	1.57
		Anti-BoAHV-5	1.22	1.40	1.85	Anti-BoAHV-5	1.18	1.19	1.43
	MVA-RecBoAHV HE	Anti-BoAHV-1	1.43	1.68	2.74	Anti-BoAHV-1	1.81	1.93	2.05
		Anti-BoAHV-5	1.93	2.49	3.90	Anti-BoAHV-5	4.18	4.80	2.98
	Inactivated BoAHV1/5	Anti-BoAHV-1	1.27	1.48	1.83	Anti-BoAHV-1	1.29	1.63	1.90
		Anti-BoAHV-5	1.25	1.41	2.00	Anti-BoAHV-5	2.16	1.86	1.90

The ELISA index was calculated as the ratio of sample OD to the cutoff, with the cutoff defined as the mean OD of negative controls plus two standard deviations. Samples with index values > 1.1 were considered positive (highlighted in red), <0.8 negative (highlighted in green), and those between 0.8 and 1.1 indeterminate. (-) indicates that experiments were not performed for these points.

## Data Availability

The raw data supporting the conclusions of this article will be made available by the authors on request.

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
