# Peer review of "A Multi-Epitope Recombinant Vaccine Candidate Against Bovine Alphaherpesvirus 1 and 5 Elicits Robust Immune Responses in Mice and Rabbits"

_vaccines, 2025, doi:10.3390/vaccines13111115_

Round 1
Reviewer 1 Report
Comments and Suggestions for Authors
The manuscript by Barbosa et al. explored a multi-epitope recombinant vaccine targeting BoAHV-1 and BoAHV-5. A synthetic gene encoding immunodominant epitopes from gB, gD, and VP8 proteins was designed and expressed in both a prokaryotic system (RecBoAHV) and a Modified Vaccinia Ankara viral vector (MVA-BoAHV). The immunogenicity of these candidates was assessed in mice and rabbits. The results showed that both RecBoAHV and MVA-RecBoAHV induced strong humoral immune responses. This is good preliminary research supporting future vaccine development. However, I still have concerns to be addressed.
Major concerns:
- The study primarily focuses on humoral responses. Cellular immune responses (e.g., CTL activity, cytokine production) are not thoroughly investigated.
- Epitope selection relies on SYFPEITHI (2007 version), which is less advanced than modern tools (e.g., IEDB). No IgG subclass analysis or neutralization assays to assess functional antibodies.
- Mice and rabbits may not fully represent the complexities of the bovine immune system. Data in cattle is needed.
Minor concerns:
- Suggest assessing the cross-reactivity of the vaccine-induced antibodies with different BoAHV-1 and BoAHV-5 strains. This will help determine the breadth of protection.
- Suggest adding a conclusion section.
- The discussion overstates implication that the recombinant vaccine candidates is a approach for controlling BoAHV-1 and BoAHV-5 infections. Further in vivo and challenge experiments are needed.
Author Response
Comments and Suggestions for Authors
The manuscript by Barbosa et al. explored a multi-epitope recombinant vaccine targeting BoAHV-1 and BoAHV-5. A synthetic gene encoding immunodominant epitopes from gB, gD, and VP8 proteins was designed and expressed in both a prokaryotic system (RecBoAHV) and a Modified Vaccinia Ankara viral vector (MVA-BoAHV). The immunogenicity of these candidates was assessed in mice and rabbits. The results showed that both RecBoAHV and MVA-RecBoAHV induced strong humoral immune responses. This is good preliminary research supporting future vaccine development. However, I still have concerns to be addressed.
We would like to thank you for the revision of the manuscript, and point out that the results even can support future vaccine development.
Major concerns:
Comments:
- The study primarily focuses on humoral responses. Cellular immune responses (e.g., CTL activity, cytokine production) are not thoroughly investigated.
- Mice and rabbits may not fully represent the complexities of the bovine immune system. Data in cattle is needed.
Response: We understand and agree with your point of view. However, we would like to explain that this manuscript, although preliminary and focused more on humoral responses at this time, had the most important objective of to present the characterization of the biotechnological tools (recombinant protein and MVA recombinant virus), and to support the proof of principle of the constructs as a vaccine candidates, what we have reached. We worked with two species models, mice and rabbits. Although mice is not complete suitable model for bovines, it is still a model that has a diversity of commercial inputs to support the experimental phase, and our research group has used this model with success to understand the role of SOCS-2 in meningoencephalitis in BoAHV-5 infection (Barbosa et al., Comp Immunol Microbiol Infec Dis, 47: 26-31, 2016). It is important to highlight that rabbits have been extensively used as animal model to understand the pathogenesis of bovine alphaherpesviruses (BoAHV-1 and BoAHV-5), including observation of virus distribution, spread of viruses in host, aspects of involvement of of non-neuronal cells, among other important findings (Machado et. al, Microbial Pathogenesis 2008 Jun;150(1-2):77-9. doi: 10.1016/j.jviromet.2008.03.008; Valera et al., J Virol Methods, 2008 Jun;150(1-2):77-9. doi: 10.1016/j.jviromet.2008.03.008). Actually, the rabbit model is extensively used even as a reliable disease model in the development of vaccines, therapeutics and molecular and cellular mechanisms underlying human diseases (Esteves et al., Exp Mol Med, 2018; 22;50(5):1-10. doi: 10.1038/s12276-018-0094-1).
Beyond this, and targeting the bovine model, we analysed the recBoAHV multiepitope protein (including gB, gD and from tegument phosphoprotein-UL47) using the bovine leukocyte antigens (BoLA), showing that the multiepitope recombinant protein had strong bind peptides. Hypothetically, results of this robust in silico tool should generate similar results during in vivo assays.
Following, we have shown that IgG and IgM from sera of bovine naturally infected with BoAHV-5 or BoAHV-1 recognized the multi-epitope recombinant protein (using as control antigen of BoAHV-1 and of BoAHV-5). This information was already inserted on the first version of the manuscript submitted on the topic 3.2 Expression and characterization of the recombinant multi-epitope protein of the results.
Comments: 2. Epitope selection relies on SYFPEITHI (2007 version), which is less advanced than modern tools (e.g., IEDB).
Response: We understand your concern. However, the construction, and experiments in mice model and in rabbit model are aligned with the time of three different students: a master's degree and two doctorates. Then, really the epitope selection was done a long time ago and we now use IEDB and others in our recent constructions. We believe that the preclinical assay shown in this manuscript points out the success of the construction, and in this way, new epitope selection in a different tool is not necessary. Another important point is that in the analyses using the last version of BoLA reported in the manuscript, strong epitope binders were shown for all four peptides, endorsing the construction for the target species.
Comments: No IgG subclass analysis or neutralization assays to assess functional antibodies.
As we informed in the final paragraph of the discussion, we are currently evaluating the recombinant multi-epitope protein (RecBoAHV) and MVA-RecBoAHV, using prime boost homologous and heterologue protocols in bovines. In this protocol in target species, we are using IgG and subclasses, IgM, neutralization assays, and ELISpot (Bovine IL-2, Bovine IL-17A, Bovine IL-4, Bovine IFNγ, and Bovine IL-8. We hope that in the new manuscript we can address all these questions regarding the humoral and cellular response in cattle.
Minor concerns:
Comments: 1. Suggest assessing the cross-reactivity of the vaccine-induced antibodies with different BoAHV-1 and BoAHV-5 strains. This will help determine the breadth of protection.
Response: Your comment is truly relevant. We do not have those samples in our laboratory to carry out the suggestion. However, we appreciate the excellent idea, which we can drive, maybe, for the next paper about results on bovine.
However, for the present manuscript, the reviewer 2 asked for one figure showing the alignment of peptides used in multiepitope construction and genotypes and strains of alphaherpesvirus. We did a new supplemental figure (Figure Supplementary 6) showing the alignment of pep1 (gB; B and T cell epitope), pep2 (gD; B cell epitope), pep3 (gD; T cell epitope), pep4 ( Tegument phosphoprotein; Tcell epitope) with BoAHV-1.1 (AJ004801.1 NCBI consortium sample - Switzerland, and KU198480.2 - Cooper strain - USA); BoAHV-1.2 (KM258880.1 - strain K22 - USA), and BoAHV-5 (5a - Y2611359.1 - strain-SV507.99 - Brazil; 5b - MW829288 strain A663 - Brazil; 5b - MZ420492 - strain 674/10 - Argentina; 5c - KY549446.1 - strain ISO 97/45 - Brazil; 5c - KY559403.2- strain-P160/96 - Brazil). We believe that this figure can at least partially respond to this question.
Legend of the figure:
Figure S-6. Alignment of synthetic peptides Pep1 (gB; B- and T-cell epitope), Pep2 (gD; B-cell epitope), Pep3 (gD; T-cell epitope), and Pep4 (tegument phosphoprotein; T-cell epitope) with representative sequences of BoAHV-1.1 (AJ004801.1 – Switzerland; KU198480.2, strain Cooper – USA), BoAHV-1.2 (KM258880.1, strain K22 – USA), and BoAHV-5 (5a: Y2611359.1, strain SV507.99 – Brazil; 5b: MW829288, strain A663 – Brazil, and MZ420492, strain 674/10 – Argentina; 5c: KY549446.1, strain ISO 97/45 – Brazil, and KY559403.2, strain P160/96 – Brazil). The alignment revealed complete identity (100%) for Pep1 across all sequences. For Pep2, 100% identity was observed with BoAHV-1.1 and BoAHV-1.2 (AJ004801.1, KU198480.2, and KM258880.1), while showed 73% identity for BoAHV-5 (Y2611359.1-5a, MW829288-5b, MZ420492-5b, KY549446.1-5c, and KY559403.2-5c). For Pep3, 100% identity was observed with BoAHV-1.1 and BoAHV-1.2 (AJ004801.1, KU198480.2, and KM258880.1), whereas 93% identity was observed with BoAHV-5 (Y2611359.1-5a, MW829288-5b, MZ420492-5b, KY549446.1-5c, and KY559403.2-5c). Pep4 exhibited 100% identity with all sequences analyzed.
Comments: 2. Suggest adding a conclusion section.
Response: We agreed and inserted a conclusion after discussion, including information about cattle’s experiments that are currently in development.
Conclusion: This vaccine candidate, based on conserved epitopes from BoAHV-1.1, BoAHV-1.2, and BoAHV-5 structural (gB, gD) and non-structural tegument phosphoprotein, ena-bled the design of a multi-epitope recombinant antigen with broad immunogenic po-tential. Predicted B-cell epitopes with high affinity for multiple BoLA haplotypes were confirmed by recognition of the recombinant protein by sera from naturally infected cattle. In pre-clinical studies, prime-boost homologous and heterologous vaccination with RecBoAHV and MVA-RecBoAHV elicited robust humoral responses, including elevated IgG in rabbits and IgG/IgM in mice against BoAHV-1 and BoAHV-5, high-lighting the vaccine’s potential. Further evaluation of cellular immunity, IgG sub-classes, neutralizing antibodies, and cytokine responses is underway in ongoing bovine studies. (lines 612-622)
Comments: 3. The discussion overstates implication that the recombinant vaccine candidates is a approach for controlling BoAHV-1 and BoAHV-5 infections. Further in vivo and challenge experiments are needed.
Response: Your comment is relevant, although we used the word “potential” and this statement was changed as bellow:
Altogether, these preliminary findings support the immunogens’ potential as a vaccine antigen candidate, targeting both BoAHV-1 and BoAHV-5. Nevertheless, clinical ex-periments in cattle, the target species, are crucial to confirm the vaccine efficacy potential. (lines 545-548)
Reviewer 2 Report
Comments and Suggestions for Authors
The full report is attached.

The manuscript is generally understandable and conveys the intended scientific message. However, several areas require linguistic refinement to enhance clarity, professionalism, and overall readability. The current version contains numerous grammatical inconsistencies, awkward phrasing, article misuse, and overly long or complex sentence constructions. These issues may hinder the flow of information and weaken the scientific tone:
-Articles are frequently omitted or misused throughout the text. For example, phrases like "in immune system" should read "in the immune system", and "use of recombinant virus" should be "the use of a recombinant virus."
-Shifts between present and past tense occasionally cause confusion. Ensure consistent use of past tense for describing completed experimental procedures and present tense for general facts or conclusions.
-Many sentences are overly long or use redundant expressions. Consider simplifying sentence constructions to improve flow. For example:
"The recombinant construct was confirmed using PCR and Western blotting which provided visual confirmation of the expression..." could be revised as: "The recombinant construct was confirmed by PCR and Western blotting, validating its expression."
-Some phrases lack precision or sound conversational. Replace casual expressions like "clearly showed" or "was seen to be increased" with more formal alternatives such as "demonstrated" or "was elevated."
-Phrases like “increased immune activation” or “good immunogenicity” should be more specific. What markers were increased? What parameters define “good”?
-Figure captions require expansion for self-sufficiency, and terminology should be standardized (e.g., define abbreviations upon first use, such as “RecBoAHV”).
-Isolated typographical and grammar errors are present (e.g., missing commas, incorrect subject–verb agreement). A few examples: “the responses was measured” should be “the responses were measured.” and “it have potential” should be “it has potential.”
While the language is sufficient to understand the scientific content, a comprehensive language revision by a fluent English speaker or a professional scientific editing service is strongly recommended before publication. This will ensure the clarity, tone, and flow of the manuscript meet the standards expected of high-impact scientific writing.
Author Response
Comments on the Quality of English Language
The manuscript is generally understandable and conveys the intended scientific message. However, several areas require linguistic refinement to enhance clarity, professionalism, and overall readability. The current version contains numerous grammatical inconsistencies, awkward phrasing, article misuse, and overly long or complex sentence constructions. These issues may hinder the flow of information and weaken the scientific tone:
-Articles are frequently omitted or misused throughout the text. For example, phrases like "in immune system" should read "in the immune system", and "use of recombinant virus" should be "the use of a recombinant virus."
-Shifts between present and past tense occasionally cause confusion. Ensure consistent use of past tense for describing completed experimental procedures and present tense for general facts or conclusions.
-Many sentences are overly long or use redundant expressions. Consider simplifying sentence constructions to improve flow. For example:
"The recombinant construct was confirmed using PCR and Western blotting which provided visual confirmation of the expression..." could be revised as: "The recombinant construct was confirmed by PCR and Western blotting, validating its expression."
-Some phrases lack precision or sound conversational. Replace casual expressions like "clearly showed" or "was seen to be increased" with more formal alternatives such as "demonstrated" or "was elevated."
-Phrases like “increased immune activation” or “good immunogenicity” should be more specific. What markers were increased? What parameters define “good”?
-Figure captions require expansion for self-sufficiency, and terminology should be standardized (e.g., define abbreviations upon first use, such as “RecBoAHV”).
-Isolated typographical and grammar errors are present (e.g., missing commas, incorrect subject–verb agreement). A few examples: “the responses was measured” should be “the responses were measured.” and “it have potential” should be “it has potential.”
While the language is sufficient to understand the scientific content, a comprehensive language revision by a fluent English speaker or a professional scientific editing service is strongly recommended before publication. This will ensure the clarity, tone, and flow of the manuscript meet the standards expected of high-impact scientific writing.
Response: We sincerely thank you for your detailed and constructive feedback regarding the quality of the English language in our manuscript. We carefully revised the entire text to address the issues you highlighted. Figure legends have also been expanded to ensure they are self-explanatory.
We agree that linguistic refinement is essential to enhance clarity and professionalism, and we have implemented substantial improvements accordingly.
The figure legends were modified as following:
Figure 1. In silico evaluation of the RecBoAHV multi-epitope construct. (A) Prediction of linear B-cell epitopes using the BepiPred 3.0 tool. The full amino acid sequence of the RecBoAHV construct was analyzed, with residues scoring above the default threshold of 0.15 considered as potential antigenic sites. Regions exceeding the threshold are shown, with a prominent peak corresponding to residues 20–45 (within Pep1), which displayed the highest B-cell epitope scores. (B) Prediction of T-cell epitopes based on binding affinity to bovine leukocyte antigen (BoLA) class I haplotypes, using the NetMHCpan 4.1 (NetMHCpan 4.1 - DTU Health Tech - Bioinformatic Services). A total of 32 peptides were analyzed across 62 BoLA haplotypes (15 in Pep1, 8 in Pep2, 6 in Pep3, and 3 in Pep4). Strong binders (SB) were defined as peptides with predicted affinity ≤ 0.5 and are indicated in the heatmap. (C) Predicted three-dimensional structure of the RecBoAHV protein, generated by I-TASSER (I-TASSER server for protein structure and function prediction). The model is shown in three orientations, with α-helices, β-strands, and loop regions annotated. Selected epitopes are mapped on the protein surface to illustrate their accessibility and structural context. (lines 330-344)
Figure 2. Characterization of RecBoAHV and reactivity with sera from naturally infected cattle. (A) Expression and purification of the RecBoAHV protein. The synthetic multi-epitope gene was cloned into the pQE-30 vector and expressed in E. coli M15 cells following IPTG induction. Protein extracts were analyzed by SDS–PAGE: (M) molecular weight marker; (UI) uninduced colony extract; (I) induced colony extract; (E1–E5) sequential fractions obtained after purification by Ni-NTA affinity chromatography. A distinct band corresponding to the expected size of ~12.7 kDa was observed in induced and purified fractions. (B) Antigenicity of RecBoAHV assessed by Western blotting using bovine sera. Membranes were probed with: (1) serum from a BoAHV-1/BoAHV-5-negative animal; (2) serum highly positive for BoAHV-1; (3) serum highly positive for BoAHV-5; and (4) serum highly positive for both BoAHV-1 and BoAHV-5. The recombinant protein was specifically recognized by positive sera, confirming the immunological relevance of the selected epitopes. (C) Serological recognition of RecBoAHV by naturally infected cattle. IgG and IgM indirect ELISA was performed using bovine sera positive for BoAHV-1 and/or BoAHV-5. Positive sera consistently reacted with the recombinant protein for both immunoglobulins, reinforcing the preservation of its antigenic determinants and supporting its potential use for immunization or serological applications. (lines 361-377)
Figure 3. Evaluation of antibody responses in New Zealand rabbits after vaccination with RecBoAHV. Indirect ELISA for anti-rabbit IgG detection using as antigens the recombinant multi-epitope protein (RecBoAHV), BoAHV-1, and BoAHV-5 purified viral proteins. Rabbits were immunized with RecBoAHV following a homologous prime-boost-boost protocol (three subcutaneous doses at 21-day intervals). Serum samples were collected at different time points (prime, boost 1, and boost 2) and analyzed for IgG reactivity. Statistically significant differences were observed between immunized and control groups already after the prime dose, with a marked increase in antibody titers following the second boost. By boost 2, high IgG levels were detected against all tested antigens, with absorbance values above 2.0. Rabbits from the control group (PBS) showed no reactivity to any of the antigens. Statistical analysis was performed using the Kruskal–Wallis followed by Dunn’s multiple comparisons test. Each group contained two animals, which were tested in three technical replicates. (lines 390-402)
Figure 4. Evaluation of antibody responses in New Zealand rabbits after vaccination with MVA-RecBoAHV. Rabbits were pre-immunized with PBS, MVA-GFP, or VACV-WR to assess the potential effect of pre-existing poxvirus immunity on the immunogenicity of MVA-RecBoAHV. Subsequently, animals were immunized with MVA-RecBoAHV using a homologous prime-boost-boost vaccination scheme. Serum samples were collected 14 days after the second boost and analyzed by indirect ELISA for IgG reactivity against BoAHV-1 and BoAHV-5 purified viral proteins. As controls, groups of rabbits immunized only with RecBoAHV protein or PBS were included. Regardless of prior exposure to MVA or VACV, animals vaccinated with MVA-RecBoAHV developed high levels of specific antibodies against both BoAHV-1 and BoAHV-5, comparable to the responses observed in the group immunized with RecBoAHV protein. In contrast, no significant reactivity was detected in the PBS control group. Statistical comparisons were performed using the Kruskal–Wallis followed by Dunn’s multiple comparisons test. The PBS, MVA-GFP, and VACV groups were evaluated in two animals/group and tested in three technical replicates. The PBS control and RecBoAHV groups were evaluated in three animals and tested in single replicates. (lines 422-437)
Figure 5. Evaluation of antibody responses in C57BL/6 mice after vaccination. Indirect ELISA anti-bovine IgG and IgM for BoAHV1 and BoAHV5 antigens from mice immunized with MVA-RecBoAHV. The animals were immunized in a prime-boost-boost protocol. All groups evaluated were tested on six animals per group. Prime and Boost 1 were evaluated in a pool format, being tested in two technical replicates. Boost 2 was evaluated individually in single replicates. The statistics presented for the boost 2 results correspond to: IgG BoAHV1 was performed using ANOVA multiple comparisons test with post-hoc Tukey’s. IgG BoAHV5 was performed using Kruskal–Wallis multiple comparisons test with post-hoc Dunn’s. IgM BoAHV5 was performed using ANOVA multiple comparisons test with post-hoc Tukey’s test. (lines 444-453)
Figure S-3. Predicted binding affinity of BoLA class I haplotypes to RecBoAHV-derived peptides (Pep1 to Pep4) using the NetMHCpan tool (version 4.1). The heatmap represents the in-silico predicted binding strength of each peptide to a panel of bovine leukocyte antigen (BoLA) class I haplotypes. Peptides are grouped according to their position within the antigen: Pep1 (gB; B- and T-cell epitope), Pep2 (gD; B-cell epitope), Pep3 (gD; T-cell epitope), and Pep4 (tegument phosphoprotein; T-cell epitope). Color intensity indicates predicted binding affinity: darker shades correspond to stronger binding, while lighter shades indicate weak or non-binding interactions. Promiscuous epitopes are those showing consistently strong binding across multiple BoLA haplotypes, highlighting peptides with broad potential immunogenicity in diverse bovine populations.
Figure S-4. Verification of the cloning of the target RecBoAHV sequence into the MVA-RecBoAHV genome by Sanger sequencing. A pair of M13 primers (Invitrogen, EUA; 10 pMol/µL), annealing to the flanking regions of the pLW44 plasmid was used for PCR amplification. The obtained sequence was aligned with the RecBoAHV reference sequence and the pLW44 plasmid sequence to confirm correct insertion of the RecBoAHV gene into the MVA genome. This analysis ensures that the recombinant virus contains the insertion of codifying region for the intended antigenic and proper plasmid flanking regions.
Figure S-5. Predicted binding affinity of HLA class I haplotypes to RecBoAHV-derived peptides (Pep1 to Pep4) using the NetMHCpan tool (version 4.1). The heatmap shows the in-silico predicted binding strength of each peptide to a panel of human leukocyte antigen (HLA) class I haplotypes. Peptides are grouped according to their position within the antigen: Pep1 (gB; B- and T-cell epitope), Pep2 (gD; B-cell epitope), Pep3 (gD; T-cell epitope), and Pep4 (tegument phosphoprotein; T-cell epitope). Color intensity reflects predicted binding: darker shades represent strong binders, while lighter shades correspond to weak or non-binders. Promiscuous epitopes with broad HLA recognition are easily identifiable as consistently strong binding across multiple haplotypes, highlighting their potential for universal immunogenicity.
Figure S-6. Alignment of synthetic peptides Pep1 (gB; B- and T-cell epitope), Pep2 (gD; B-cell epitope), Pep3 (gD; T-cell epitope), and Pep4 (tegument phosphoprotein; T-cell epitope) with representative sequences of BoAHV-1.1 (AJ004801.1 – Switzerland; KU198480.2, strain Cooper – USA), BoAHV-1.2 (KM258880.1, strain K22 – USA), and BoAHV-5 (5a: Y2611359.1, strain SV507.99 – Brazil; 5b: MW829288, strain A663 – Brazil, and MZ420492, strain 674/10 – Argentina; 5c: KY549446.1, strain ISO 97/45 – Brazil, and KY559403.2, strain P160/96 – Brazil). The alignment revealed complete identity (100%) for Pep1 across all sequences. For Pep2, 100% identity was observed with BoAHV-1.1 and BoAHV-1.2 (AJ004801.1, KU198480.2, and KM258880.1), while showed 73% identity for BoAHV-5 (Y2611359.1-5a, MW829288-5b, MZ420492-5b, KY549446.1-5c, and KY559403.2-5c). For Pep3, 100% identity was observed with BoAHV-1.1 and BoAHV-1.2 (AJ004801.1, KU198480.2, and KM258880.1), whereas 93% identity was observed with BoAHV-5 (Y2611359.1-5a, MW829288-5b, MZ420492-5b, KY549446.1-5c, and KY559403.2-5c). Pep4 exhibited 100% identity with all sequences analyzed.
Comments: The study encompasses in silico epitope prediction (BoLA and HLA binding affinities), construction and confirmation of recombinant virus using MVA vector, and assessment of immunogenicity in murine and rabbit models. The vaccine candidate elicited antigen-specific humoral and cellular responses, supporting its potential as a cross-protective platform against BoHV-1 and BoHV-5. The study is original, technically sound, and provides meaningful insights into veterinary vaccinology. The manuscript addresses a critical gap in veterinary vaccine development, particularly for co-endemic BoHV-1 and BoHV-5, which remain major causes of respiratory and neurological disease in cattle. The use of a multi-epitope approach, coupled with MVA-based expression and dual-species preclinical testing, is well-conceived and appropriate. The combination of in silico, molecular biology, and immunological methods supports the multidimensional nature of the study. Inclusion of both murine and rabbit models provides a broader assessment of immunogenicity across species. The supplementary material (Figures Sl— S3) strengthens the in silico rationale and provides visual confirmation of epitope selection and vector construction. However, there are areas requiring clarification and refinement before publication:
Response: We sincerely thank you for the thoughtful and encouraging assessment of our manuscript. We greatly appreciate your recognition of the originality, technical soundness, and translational potential of our study, as well as the multidimensional approach combining in silico, molecular, and immunological methods. Your acknowledgment of the importance of addressing co-endemic BoAHV-1 and BoAHV-5 and of including both murine and rabbit models to broaden the immunogenicity assessment is also highly valued. We carefully considered all your points and have revised the manuscript accordingly to provide the requested clarifications and refinements.
Comments: 1. While immunogenicity is clearly demonstrated, no challenge experiments were performed to validate actual protective efficacy in vivo. Although the authors acknowledge this limitation in the discussion, a clearer elaboration on how these results translate to protective potential, or a mention of planned future experiments, would enhance impact.
Response: We thank the reviewer for raising this important point regarding the absence of challenge experiments. We fully acknowledge that assessing protection by viral challenge would provide direct evidence of efficacy. However, such experiments are technically demanding and require high biosafety level facilities, and are not accessible to our group at this stage. For this reason, our present work was focused on the immunogenicity profile of the vaccine candidate in preclinical models. Importantly, we are already conducting studies in the target species (cattle), evaluating homologous and heterologous booster protocols in two age groups (calves and heifers). These studies include the monitoring of vaccinated dams to assess the transfer of immunity to their offspring. We believe this stepwise approach, starting with robust immunogenicity evaluation in different models, followed by targeted studies in cattle, represents a scientifically sound and ethically responsible strategy for advancing this vaccine candidate.
Additionally, we have revised the discussion and conclusion to explicitly acknowledge the limitations of this study.
Discussion: Although specific cellular responses remain to be fully characterized, our data suggest that the combination of recombinant multi-epitope protein and a viral vector is a promising strategy for inducing broad and potentially effective immunity. The inclusion of appropriate adjuvants may enhance immunogenicity and broaden the protective potential of the formulation, representing a promising path for practical applications in cattle immunization. Future investigations should address these gaps, including challenge studies with the virus in relevant animal models. Although protection was not directly assessed via viral challenge in the present study, the observed immunogenic profile supports the rationale for conducting further efficacy trials in cattle. Indeed, a study of this vaccine’s immunogenicity and efficacy in bovines is currently being conducted. A further limitation of the present study is the lack of virus neutralization assays, which are essential to evaluate the functional capacity of vaccine-induced antibodies. To address this, virus neutralization assays are currently being performed in cattle, and results will provide critical complementary data to the immunogenicity findings reported here (lines 598-611).
- Conclusion
This vaccine candidate, based on conserved epitopes from BoAHV-1.1, BoAHV-1.2, and BoAHV-5 structural (gB, gD) and non-structural tegument phosphoprotein, ena-bled the design of a multi-epitope recombinant antigen with broad immunogenic po-tential. Predicted B-cell epitopes with high affinity for multiple BoLA haplotypes were confirmed by recognition of the recombinant protein by sera from naturally infected cattle. In pre-clinical studies, prime-boost homologous and heterologous vaccination with RecBoAHV and MVA-RecBoAHV elicited robust humoral responses, including elevated IgG in rabbits and IgG/IgM in mice against BoAHV-1 and BoAHV-5, high-lighting the vaccine’s potential. Further evaluation of cellular immunity, IgG sub-classes, neutralizing antibodies, and cytokine responses is underway in ongoing bovine studies (lines 612-622).
Comments: 2. Since this vaccine targets both BoHV-1 and BoHV-5, more detailed discussion on conserved versus variable epitopes across these viruses is warranted. Are the selected epitopes truly conserved across genotypes and strains? A short alignment figure or table in supplementary material could improve this.
Response: We thank you for this important suggestion. In response, we have included a short discussion on conserved versus variable epitopes across BoAHV‑1 and BoAHV‑5 in the Discussion section. Additionally, we have provided a sequence alignment of the four synthetic peptides with representative BoAHV‑1 and BoAHV‑5 sequences as Supplementary Figure S6, which illustrates the conservation and variability of these epitopes across different strains. We believe that these additions address the reviewer’s comment and strengthen the manuscript by providing a clearer rationale for the selection of cross-protective epitopes.
Discussion: These observations are consistent with the general genomic stability observed in BoAHV 1 and BoAHV 5. Studies of whole-genome sequencing have shown that essential structural genes, such as gB, gC, gD, gH, and gM, are highly conserved across viral strains, whereas most genetic variation tends to accumulate in non-essential or surface-exposed regions of glycoproteins, potentially contributing to some antigenic diversity [45-47]. This overall pattern is consistent with the alignment results presented here, in which epitopes derived from gB (Pep1) and tegument phosphoprotein (Pep4) were fully conserved, while gD-derived epitopes (Pep2 and Pep3) showed strain- or genotype-associated divergence, particularly in BoAHV 5 sequences. The inclusion of these conserved epitopes in vaccine design is, therefore, expected to enhance cross-protective immunity (see Figure S-6 for alignment of peptides with different genotypes of BoAHV-1 and BoAHV-5). (lines 528-538).
Figure S-6. Alignment of synthetic peptides Pep1 (gB; B- and T-cell epitope), Pep2 (gD; B-cell epitope), Pep3 (gD; T-cell epitope), and Pep4 (tegument phosphoprotein; T-cell epitope) with representative sequences of BoAHV-1.1 (AJ004801.1 – Switzerland; KU198480.2, strain Cooper – USA), BoAHV-1.2 (KM258880.1, strain K22 – USA), and BoAHV-5 (5a: Y2611359.1, strain SV507.99 – Brazil; 5b: MW829288, strain A663 – Brazil, and MZ420492, strain 674/10 – Argentina; 5c: KY549446.1, strain ISO 97/45 – Brazil, and KY559403.2, strain P160/96 – Brazil). The alignment revealed complete identity (100%) for Pep1 across all sequences. For Pep2, 100% identity was observed with BoAHV-1.1 and BoAHV-1.2 (AJ004801.1, KU198480.2, and KM258880.1), while showed 73% identity for BoAHV-5 (Y2611359.1-5a, MW829288-5b, MZ420492-5b, KY549446.1-5c, and KY559403.2-5c). For Pep3, 100% identity was observed with BoAHV-1.1 and BoAHV-1.2 (AJ004801.1, KU198480.2, and KM258880.1), whereas 93% identity was observed with BoAHV-5 (Y2611359.1-5a, MW829288-5b, MZ420492-5b, KY549446.1-5c, and KY559403.2-5c). Pep4 exhibited 100% identity with all sequences analyzed.
Comments: 3. While ELISA-based IgG detection is shown, there is no report of virus neutralization assays. This limits conclusions about vaccine efficacy. At a minimum, the authors should acknowledge this in the discussion as a limitation.
Response: We appreciate your comment regarding the absence of virus neutralization assays. We agree that virus neutralization is a critical parameter to evaluate the protective efficacy of a vaccine candidate. As suggested, we have now acknowledged this as a limitation in the Discussion section. Additionally, we highlight that neutralization assays are currently being performed in the target species (cattle), which will provide essential data to complement the immunogenicity findings reported here.
Discussion: Although specific cellular responses remain to be fully characterized, the data suggest that the combination of the multi-epitope protein with the viral vector is a promising strategy for inducing broad and potentially effective humoral immunity. The combination with appropriate adjuvants may enhance immunogenicity and broaden the protective potential of the formulation, representing a promising path for practical applications in cattle immunization. Future investigations should address these gaps, including challenge studies with the virus in relevant animal models. Although protection was not directly assessed via viral challenge in the present study, the observed immunogenic profile supports the rationale for conducting further efficacy trials in a bovine model. A study of the formulations in the target species (cattle) is currently being conducted by our research group. A further limitation of the present study is the lack of virus neutralization assays, which are essential to evaluate the functional capacity of vaccine-induced antibodies. To address this, virus neutralization assays are currently being performed in cattle, and their results will provide critical complementary data to the immunogenicity findings reported here. (lines 599-612)
Comments: 4. It is not clear whether the experiments were statistically powered. The number of animals per group should be explicitly stated in the main figures and methods. Include a short statement on how the group sizes were determined or justified (3Rs, previous studies, etc.).
Response: We planned the experiments according to the premise that calculation of the required sampling (number of animals for a given experimental group) is related to the confidence interval of the mean (CI). Very unstable variables (greater deviations) will have less credibility in their mean, unless r (number of repetitions, animals) is increased. Little unstable variables, on the other hand, will not require a high value of r, as they inherently have a low value of s (deviation). Then, the number of animals per group was determined after reviewing published scientific literature about vaccinal experiments for bovine alphaherpesviruses in peer-reviewed journals. This direction ensured that the results are significant, so there would be no need for repetitions, thus guaranteeing the necessary practice of reducing the 3Rs. Statistical analyses were performed using the GraphPad Prism 8.0.1 program, employing one-way ANOVA with Tukey post-test for parametric analyses or Kruskal-Wallis followed by non-parametric analyses. Significance levels were set at p<0.05. (Literature used to write the ethical protocols submitted 1.SAMPAIO, I.B.M. Estatística aplicada à Experimentação Animal. Belo Horizonte: Fundação de. Ensino e Pesquisa em Medicina Veterinária e Zootecnia, 1998; 2.ARMITAGE, P. & Berry, G. Statistical Methods in Medical Research. London: Blackwell Science ltd, 1995. 620p; 3.SAMPAIO, I.B.M. Estatística Aplicada à Experimentação Animal. FEP-MVZ, Belo Horizonte, 2002. 265p; 4. SNEDECOR G.W & Cochran, W.G. Statistical Methods. Iowa State Univ. Press, Ames, 1980; 5.STEEL, R.G.D. & Torrie, J.H. Principles and Procedures of Statistics. McGraw-Hill, New York,1980).
The information was indeed missing. Thank you.
The UFMG-CEUA follows all IACUC guidelines to minimize animal suffering. Group sizes were chosen after review of the literature on bovine alphaherpesvirus preclinical studies and in-house pilot data. Several published BoAHV studies use 6 mice/group per group for immunogenicity analyses [16-18] and small rabbit cohorts (2 animals/group) for preliminary immunogenicity and protection tests [19,20]. These sample sizes balance statistical sensitivity with the 3Rs principle of reduction and the stepwise approach from pilot testing to larger confirmatory trials. (lines 97-103)
The rabbits were divided into two groups (two animals/group) and immunized in a prime-boost-boost homologue protocol consisting of three subcutaneous doses at 21-day intervals. (lines 186-188)
For the evaluation of the MVA-RecBoAHV construct, animals were divided into three groups, with two animals per group. (197-199)
[...] PBS control (unvaccinated) and RecBoAHV-immunized groups (three animals/group). (lines 201-202)
Figure 3. [...] Each group contained two animals, which were tested in three technical replicates. (lines 401-402).
Figure 4. [...] The PBS, MVA-GFP, and VACV groups were evaluated in two animals/groups and tested in three replicates. The PBS control and RecBoAHV groups were evaluated in three animals and tested in single replicates. (lines 434-437).
Figure 5. [...] All groups evaluated were tested on six animals per group. Prime and Boost 1 were evaluated in a pool format, being tested in two replicates. Boost 2 was evaluated individually in single replicates. (lines 446-449)
We would like to inform, also, that the data presented in this manuscript used two ethical protocols at Universidade Federal de Minas Gerais - CETEA - 239/2011, approved on 07.11.2012 and CEUA - 350/2018 approved on 25.02.2019. Indeed, we forgot to insert the last one of 2018 and we would like to apologise for this mistake. Both acronyms refer to the Ethics Committee on the Use of Animals, CETEA is older and CEUA is the more recent acronym.
Institutional Review Board Statement: The animal study protocol was approved by Universidade Federal de Minas Gerais - Ethics Committee on the Use of Animals: CETEA - 239/2011, approved on 07.11.2012, and CEUA - 350/2018, approved on 25.02.2019. (lines 664-666)
Comments: 5. The supplementary figures are useful but should be referenced more clearly in the main text (e.g., “see Fig. S-1 for BoLA binding affinities”).
Response: We thank you for this valuable suggestion and agree that including this information will improve the clarity and understanding of the results.
(see Figure S-3 for BoLA strong or weak binding affinities) (lines 305-306)
(see Figure S-4 for alignment of the MVA-RecBoAHV construct with the RecBoAHV gene) (lines 408-409)
(see Figure S-5 for MHC haplotypes strong and weak binding in mice) (lies 458-459)
Comments: 6. Line 75-80: the rationale for choosing these epitopes (Pepl—Pep4) could benefit from clearer justification, including whether these were selected based on sequence conservation, immunodominance, or structural exposure.
Response: We thank the reviewer for this valuable comment. The rationale for selecting the four epitopes (Pep1–Pep4) was based on three complementary criteria: (i) sequence conservation across BoAHV-1 and BoAHV-5 strains, (ii) structural exposure/accessibility of the target proteins, and (iii) their immunological relevance in the context of BoAHV infection, particularly regarding their ability to induce T- and/or B-cell responses. The sentence below has been added to the results section.
These proteins were chosen not only because they are conserved among BoAHV1 and -5, but also for their well-documented immunological relevance in BoAHV infection. Glycoprotein B (gB) and glycoprotein D (gD) are essential for viral adsorption, entry, and cell-to-cell spread, and have been shown to elicit robust humoral and cellular immune responses, including the induction of neutralizing antibodies and CD4+/CD8+ T-cell activation. The tegument phosphoprotein, although less accessible to antibody recognition, is abundant in virions and capable of inducing strong T-cell responses, contributing to effective cellular immunity [22,23]. (lines 270-278)
Comments: 7. Line 124: please clarify whether the MVA backbone used has deletions in E3L or other virulence genes that affect immunogenicity or attenuation.
Response: Thank you for your observation. The information was missing. We have included the information in section 2.3 of the Methodology.
The construction of the recombinant vaccine vector was based on the homologous recombination between the transfer plasmid. Cells infected with recombinant vector clones were selected with the aid of GFP expression, and clones expressing MVA-RecBoAHV were subsequently isolated. (lines 166-169)
Comments: 8. Line 190—192: consider presenting a table comparing antibody titers across time points in both species to better illustrate immunological kinetics.
Response: We appreciate your suggestion, and believe it enriched the results.
Methodology: The ELISA index was calculated as the ratio of sample OD to the cutoff, with the cutoff defined as the mean OD of negative controls plus two standard deviations. Samples with index values >1.1 were considered positive (highlighted in red), <0.8 negative (highlighted in green), and those between 0.8 and 1.1 indeterminate. (lines 256-260)
Results: Comparative analysis of ELISA indices for IgG and IgM revealed distinct responses across the animal models and immunization regimens. In rabbits immunized with RecBoAHV, robust IgG responses were observed against BoAHV-1 and BoAHV-5, with indices reaching 7.89 and 6.22, respectively, after the second boost. In contrast, rabbits vaccinated with MVA-RecBoAHV HO exhibited more moderate responses, with indices of 2.94 against BoAHV-1 and 5.39 against BoAHV-5 following the second boost.
In the mice experiments, all immunizing agents induced seroconversion for both IgG and IgM, with indices above the cutoff as early as the priming dose. Animals immunized with RecBoAHV exhibited a progressive increase in IgG and IgM indices against both viruses, reaching up to 1.75 (IgG anti-BoAHV-5) and 1.72 (IgM anti-BoAHV-1) after the second boost. Vaccination with MVA-RecBoAHV HO (homologous protocol) induced moderate responses, with IgG indices ranging from 1.11 to 1.85 and IgM indices from 1.19 to 1.57. MVA-RecBoAHV HE (heterologous protocol) elicited the highest serological responses, particularly for IgG anti-BoAHV-5 (3.90) and IgM anti-BoAHV-5 (4.80) after the first boost. The inactivated BoAHV-1/5 vaccine promoted intermediate levels of IgG (up to 2.00) and IgM (up to 2.16) against both viruses.
Table 3. Comparative IgG and IgM ELISA indices between animal models and immunizing conditions.
The ELISA index was calculated as the ratio of sample OD to the cutoff, with the cutoff defined as the mean OD of negative controls plus two standard deviations. Samples with index values >1.1 were considered positive (highlighted in red), <0.8 negative (highlighted in green), and those between 0.8 and 1.1 indeterminate. (lines 473-495)
Comments: 9. Line 250: provide details on the cytokine detection platform used (e.g., ELISA kit manufacturer, sensitivity limits, etc.).
Response: The ELISA used in this study is an in-house method developed in our laboratory and was previously described by our research group in Mansur et al., 2009. The reference has been included in the Methods section.
Mansur HS, Palhares RM, Andrade GI, Piscitelli Mansur AA, Barbosa-Stancioli EF. Improvement of viral recombinant protein-based immunoassays using nanostructured hybrids as solid support. J Mater Sci Mater Med. 2009 Feb;20(2):513-9. doi: 10.1007/s10856-008-3606-z. Epub 2008 Oct 14. PMID: 18853236.
An indirect in-house ELISA [32] was performed to evaluate the recognition of the recombinant multi-epitope protein by IgG and IgM […] (lines 222-223)
Comments: 10. Discussion, final paragraph: add a sentence explicitly acknowledging that although protection was not assessed via viral challenge, the vaccine's immunogenic profile justifies further efficacy trials in a bovine model.
Response: Thanks for the suggestion. This information was included at the end of the discussion.
Although protection was not directly assessed via viral challenge in the present study, the observed immunogenic profile supports the rationale for conducting further efficacy trials in cattle. (lines 604-605)
Comments: The manuscript presents a compelling, well-structured study with meaningful translational potential. However, the authors should address the comments above, particularly clarifying limitations, improving language and figure integration, and enhancing the discussion of cross-protection. With these refinements, the paper will be suitable for publication.
Response: We sincerely thank the reviewer for the positive evaluation of our manuscript and for recognizing its translational potential. We also appreciate the constructive feedback provided. All comments have been carefully addressed, including clarifications of study limitations, improvements to language and figure integration, and an expanded discussion of cross-protection. We believe that these revisions have strengthened the manuscript and have incorporated the reviewer’s suggestions to enhance clarity and scientific rigor.
Reviewer 3 Report
Comments and Suggestions for Authors
Dear authors,
Your manuscript “A multi-epitope recombinant vaccine candidate against Bovine alphaherpesvirus 1 and 5 elicits robust immune responses in mice and rabbits” describes the evaluation of multi-epitope recombinant vaccine candidates against Bovine alphaherpesvirus 1 and 5.
The manuscript needs a serious revision and could not be published In current form.
Some English correction is needed.
Affiliations section - is there an equivalent of the name of the lab and institute in Brazil in English? If so, please replace.
L10 – mateuslaguardia@gmail.com, amanda231182@gmail.com; - please correct to mateuslaguardia@gmail.com; amanda231182@gmail.com;
L19 – “(BoAHV-1 and BoAHV-5)” – please change to “(BoAHV-1 and BoAHV-5), respectively”;
L29 - “NetMHCpan” – should be stated. Is it software?
Keywords section - please make in the equal style - either capitalize keywords or lowercase them;
L50 – Family and subfamily names should be in italic;
L56 – “meningoencephalitis.[6].” – correct, please;
L74, 116, 122, 278 – “in silico” – all Latin words should be in italic;
L81, 133, 285 – “Escherichia coli” – should be in italic as species name;
L89 – “Oryctolagus cuniculus” – should be in italic as species name;
L90 – “Federal University of Minas Gerais” – the country should be stated;
L112 – 5’- and 3’- ends should be specifies for each primer;
L131-132 – “The synthetic gene encoding the recombinant protein was amplified from the commercially obtained plasmid pEN08H.” - It is not clear what recombinant gene/protein we are talking about here, if the one you predicted, where did this plasmid come from, etc.? Please describe it more clearly and accurately;
Section 2.4 – the expression plasmids should be decribed in short;
L134 – “induction with IPTG” – please give more information about conditions; the same for cultivation conditions;
L135 – “10 mM Tris-Cl” – correct to “10 mM Tris-HCl”;
L136 – “containing a protease inhibitor cocktail” – what do you mean? Should be stated;
L139 – “HisTrap column” – the manufacturer should be stated;
L141 – “The viral vector” – what vector? Should be described; the conditions of cultivation and titration should be described;
L143-144 – “The nucleotide sequence of the RecBoAHV multi-epitope protein, cloned in vector pGEMT-easy,” – the cloning should be described; moreover, the sequence can not be “subcloned”, but DNA fragment. Correct;
L147-148 – “were selected with 147 the aid of GFP expression” - rephrase please;
L149 – “validated through DNA sequencing” – what sequencing method and primers were used? Should be stated; “detection of transcripts” – what method was used?
L161, 179 – “ad libitum” – all Latin words should be in italic;
L166 – “phosphate buffer solution as a negative control” – what about 5% aluminum hydroxide and 25% Emulsigen in negative control? It seems that this could be more correct;
L173 – “MVA-GFP, and group 3 received inactivated VACV WR” – the doses should be stated;
Section 2.7 – the description of dose evaluation should be given;
L193 – “tween-20” – please change to “Tween-20”;
Section 2.8 –How did you prove the specificity of your in-house test system before experiments? It should be described. In other words, if you got the “wrong” recombinant protein and immunized animals with it, their sera will detect this wrong protein adsorbed onto the plates, but will have no relation to the natural intact protein.
L230 – “Hydrophobicity profiling and transmembrane domain prediction” –why this data is not presented?
General remarks – the schemes of plasmids containing recombinant DNA fragments with epitopes would be very illustrative and useful;
Figure 5 – 1) Standard deviation is not shown on the charts, why? Have you done statistical processing of the data? 2) figure’s legend is too large;
L495 – “Mycobacterium tuberculosis” – should be in italic as species name;
References section – all references should be given in equal style; please check carefully and correct.
Author Response
Comments and Suggestions for Authors
Dear authors,
Your manuscript “A multi-epitope recombinant vaccine candidate against Bovine alphaherpesvirus 1 and 5 elicits robust immune responses in mice and rabbits” describes the evaluation of multi-epitope recombinant vaccine candidates against Bovine alphaherpesvirus 1 and 5.
The manuscript needs a serious revision and could not be published In current form.
Response: We acknowledge your evaluation of our manuscript and appreciate your honest feedback. We understand that, in its current form, the work requires substantial revisions, and we are committed to addressing the issues raised in detail. We will carefully revise the manuscript to improve its clarity, rigor, and presentation, ensuring that it meets the standards required for publication.
Some English correction is needed.
Response: The manuscript has been carefully revised for English language, grammar, and clarity throughout to ensure readability and scientific accuracy.
Comments: Affiliations section - is there an equivalent of the name of the lab and institute in Brazil in English? If so, please replace.
Response: We thank the reviewer for the suggestion. According to our institution’s official guidelines, the name of the laboratory and institute must be maintained in Portuguese.
Comments: L10 – mateuslaguardia@gmail.com, amanda231182@gmail.com; - please correct to mateuslaguardia@gmail.com; amanda231182@gmail.com; (ponto e vírgula)
Response: Thanks for the correction. It has been adjusted.
Comments: L19 – “(BoAHV-1 and BoAHV-5)” – please change to “(BoAHV-1 and BoAHV-5), respectively”;
Response: Thank you. The word was added.
(BoAHV-1 and BoAHV-5), respectively, (line 19)
Comments: L29 - “NetMHCpan” – should be stated. Is it software?
Response: Thank you. The correction has been added to the abstract.
NetMHCpan tool (version 4.1). (line 29)
Comments: Keywords section - please make in the equal style - either capitalize keywords or lowercase them;
Response: We thank the reviewer for this suggestion. The keywords have been standardized to use sentence case consistently throughout the manuscript:
Keywords: Bovine alphaherpesvirus; BoAHV-1; BoAHV-5; Recombinant vaccine; Multi-epitope antigen; MVA vector. (lines 43-44).
Comments:
L50 – Family and subfamily names should be in italic;
L56 – “meningoencephalitis.[6].” – correct, please;
L74, 116, 122, 278 – “in silico” – all Latin words should be in italic;
L81, 133, 285 – “Escherichia coli” – should be in italic as species name;
L89 – “Oryctolagus cuniculus” – should be in italic as species name;
Thank you for the correction, and we apologize for the oversight. It occurred after we transferred the manuscript for the journal template and unfortunately, we did not notice the modification inserted. All italics have been added.
L90 – “Federal University of Minas Gerais” – the country should be stated;
Response: We appreciate it. The information has been added.
Universidade Federal de Minas Gerais - Brazil (line 92)
Comments: L112 – 5’- and 3’- ends should be specified for each primer;
Response: Thanks for the information. We added the observation.
(5’ ATGGATCCCACCGCGAGCACACC 3’ and 5’ ATAAGCTTGTCGCTGCTATCGCCGTC 3’, respectively) (lines 121-123)
Comments: L131-132 – “The synthetic gene encoding the recombinant protein was amplified from the commercially obtained plasmid pEN08H.” - It is not clear what recombinant gene/protein we are talking about here, if the one you predicted, where did this plasmid come from, etc.? Please describe it more clearly and accurately;
Section 2.4 – the expression plasmids should be described in short;
L134 – “induction with IPTG” – please give more information about conditions; the same for cultivation conditions; (OK)
L135 – “10 mM Tris-Cl” – correct to “10 mM Tris-HCl”;
L136 – “containing a protease inhibitor cocktail” – what do you mean? Should be stated;
L139 – “HisTrap column” – the manufacturer should be stated;
Response: We appreciate all suggestions. Section 2.4 has been reworded to better understand the methodology used.
2.4 Cloning of the synthetic gene RecBoAHV for expression in a prokaryotic vector
The synthetic gene encoding the recombinant protein from the pGEMT-easy vector and the expression plasmid pQE-30 (Genscript, USA) were restricted with BamHI and HINDIII enzymes (Promega, USA). The DNA fragment of the RecBoAHV mul-ti-epitope protein was purified and cloned into pQE-30. The pQE-30-RecBoAHV plas-mid contains a T5 promoter for IPTG-inducible expression and an N-terminal 6×His tag for affinity purification, and was used to transform the competent E. coli M15 strain (see Figure S-1 for plasmid constructions). Protein expression was induced with 1 mM isopropyl-β-D-1-thiogalactopyranoside (IPTG) for 5 h at 37 °C with shaking at 200 rpm. Following induction, bacterial cells were harvested, and the resulting pellet was lysed in Buffer A (30 mM imidazole, 8 M urea, 10 mM Tris-HCl, 100 mM NaH₂PO₄, pH 8.0) containing a protease inhibitor cocktail [5 mM Dithiothreitol (DTT) and 1 mM Phenylmethylsulphonyl fluoride (PMSF)]. The lysate was centrifuged at 6,000 × g for 1 h at 4 °C, and the supernatant was subjected to affinity chromatography using an ÄKTA Start system (GE Healthcare Life Sciences, United Kingdom) equipped with a HisTrap column (Cytiva, USA). Purified fractions were collected and analyzed by SDS–PAGE. (lines 140-155)
Comments: L141 – “The viral vector” – what vector? Should be described; the conditions of cultivation and titration should be described;
L147-148 – “were selected with 147 the aid of GFP expression” - rephrase please;
L149 – “validated through DNA sequencing” – what sequencing method and primers were used? Should be stated; “detection of transcripts” – what method was used?
Response: We thank you for this suggestion. Section 2.5 has been reworded to better understand the methodology used.
2.5 Construction of the Recombinant MVA-RecBoAHV Vector
The MVA-RecBoAHV viral vector was selected, amplified, and titrated in BHK-21 cells (Baby Hamster Kidney fibroblasts, obtained from the American Type Culture Collection - ATCC® CCL-10™). The DNA fragment encoding the RecBoAHV multi-epitope protein was inserted into the pLW44 transfer plasmid using T/A cloning and confirmed by sequencing (see Figure S-1 for plasmid constructions). The pLW44 transfer plasmid also contains the green fluorescent protein (GFP) coding sequence under the control of the mH5 early/late Orthopoxvirus vaccinia (VACV) promoter [26]. BHK-21 cells were infected with MVA and subsequently transfected with pLW44-RecBoAHV. The MVA used in these constructions is derived from the MVA-1974 clone, kindly provided by Dr. Bernard Moss (LVD/NIAID/NIH). This low-passaged MVA was transferred to UFMG under a specific material transfer agreement (MTA). The construction of the recombinant vaccine vector was based on the homologous recombination between the transfer plasmid. Cells infected with recombinant vector clones were selected with the aid of GFP expression, and clones expressing MVA-RecBoAHV were subsequently isolated. The MVA-RecBoAHV construct was validated by Sanger sequencing using the BigDye® Terminator v1.1 Cycle Sequencing Kit (Applied Biosystems, USA) on a MegaBACE™ 1000 capillary sequencer (GE Healthcare, UK) with primers annealing to the pLW44 transfer plasmid flanking regions. Sequence data were analyzed using Chromas v2.23 and aligned with reference sequences in GenBank using BLASTn. Transcript expression in infected BHK-21 cells was detected by RT-PCR using oligo(dT) and BoAHV-specific primers. For the viral stock production, cells were infected at 37ºC, 5% CO2 atmosphere for 1 hour. After adsorption, the cells were incu-bated in the same atmosphere in DMEM (Dulbecco's Modified Eagle Medium – Sigma Aldrich, USA) supplemented with 7.5% NaHCO3, antibiotics (100 μg/mL streptomycin and 100 U/mL penicillin), antifungal (fungizone at 25 μg/mL) and 5% fetal bovine se-rum (FBS - Gibco, USA) for 48 hours. Recombinant virus purification was performed in 36% (w/v) sucrose cushion (in 10 mM Tris-HCl, pH 9.0), and the titer was determined by a plaque-based assay [27,28]. (lines 156-183)
Comments: L161, 179 – “ad libitum” – all Latin words should be in italic;
Response: Thanks for the correction, italics have been inserted. (line 186 e 197)
Comments: L166 – “phosphate buffer solution as a negative control” – what about 5% aluminum hydroxide and 25% Emulsigen in negative control? It seems that this could be more correct;
Response: We agree with the point highlighted. Unfortunately, the PBS was the negative control used at that time. In all the subsequent experiments in cattle (now in course) the control group was vaccinated with the adjuvant.
Comments: L173 – “MVA-GFP, and group 3 received inactivated VACV WR” – the doses should be stated;
Response: We appreciate your observation. The doses have been added.
[...] 1.0 × 107 PFU MVA-GFP, and Group 3 received 1.0 × 107 PFU inactivated VACV WR. (line 199-200)
Comments: Section 2.7 – the description of dose evaluation should be given;
Response: We added the following information:
MVA immunizations were defined at a dose of 1 × 10⁷ PFU, in accordance with several previous studies in mice [29-31]. Protein immunization was defined as 1.0 µg/dose. The dose choice was based on the small size of the chimeric protein (~100 amino acids), providing an adequate molar ratio of protein molecules per dose. According to the UCFS office Research (University of California San Francisco), the dose recommended for intranasal instillation of mice is 50ul maximum volume, and following this recommendation, we prepared individual doses of 10ul (divided equally between both nostrils). (lines 210-217)
Comments: L193 – “tween-20” – please change to “Tween-20”;
Response: The correction was made.
0.05% Tween-20 in Phosphate Buffered Saline pH 7.2 (lines 229)
Comments: Section 2.8 –How did you prove the specificity of your in-house test system before experiments? It should be described. In other words, if you got the “wrong” recombinant protein and immunized animals with it, their sera will detect this wrong protein adsorbed onto the plates, but will have no relation to the natural intact protein.
Response: According to Figure 2 (B), the characterization of RecBoAHV was initially accessed by analysing the reactivity with sera from naturally BoAHV-1 and BoAHV-5 infected cattle in Western blotting. Besides recombinant protein, viral antigens of both viruses were used and the results are unequivocal. Serological recognition of RecBoAHV by naturally infected cattle was done after, using IgG and IgM in indirect ELISA with bovine sera positive for BoAHV-1 and/or BoAHV-5. Another important point is about the sequencing realized after cloning procedures, proving the insertion of a region codifying the multi-epitope protein, with an expression of a protein in the right size expected (2A).
Comments: L230 – “Hydrophobicity profiling and transmembrane domain prediction” –why this data is not presented?
Response: We did the prediction in the initial experiments of recombinant construction. We will insert as below.
Hydrophobicity profiling and transmembrane domain prediction (see Figure S-2 for analysis of the hydrophobicity and transmembrane region) (lines 279-280)
Figure S-2. Computation of hydropathicity and transmembrane prediction. (A) Kyte-Doolittle Hydropathy plot (http://web.expasy.org/protparam/). (B) Transmembrane segment prediction performed using DAS-Transmembrane Prediction server (tmdas.bioinfo.se). The position of the four peptides (Pep 1 to Pep 4) that compounds the multi-epitope RecBoAHV protein is pointed out in both graphics. The Kyte Doolittle transmembrane prediction has a cut-off above 1.6, showing in (A) the absence of a transmembrane segment. In DAS prediction (B) the strict cut off (2.2) showed a small portion of the transmembrane region initiating after the end of Pep 2 and finishing soon after the start of Pep 3.
Comments: General remarks – the schemes of plasmids containing recombinant DNA fragments with epitopes would be very illustrative and useful;
Response: We agree, once we have worked with four different constructions, then the schematic figure can facilitate the comprehension. I would like to inform you that after we did the figure, we adjusted the text in the topics where the description was inserted. The text modified was written in red, as in the modification asked by the referees.
Figure S-1: Plasmid constructions with the RecBoAHV gene. (A) The synthetic gene encoding a recombinant BoAHV-1 and BoAHV-5 multi-epitope flexible protein (RecBoAHV) was custom-synthesized by Enthelecon (pEN08H, Germany). Primers for direct PCR amplification and insertion of BamHI and HindIII restriction enzyme sites to the 5’ and 3’ ends of the construct were synthesized by Integrated DNA Technologies (USA). (B) Purified PCR insert (gene coding the RecBoAHV) was subcloned into the pGEMT-easy vector system (Promega-USA). (C) The synthetic gene encoding the recombinant protein from plasmid pGEMT-easy vector and the expression plasmid pQE-30 (QIAGEN; USA) were restricted with BamHI and HindIII enzymes (PROMEGA – USA). The RecBoAHV coding gene was purified and subcloned into pQE-30 and used to transform the competent E. coli M15 strain for expression of the recombinant protein (RecBoAHV). (D) For the construction of the MVA-RecBoAHV Vector, the nucleotide sequence of the RecBoAHV multi-epitope protein cloned in vector pGEMT-easy was subcloned into pLW44 transfer plasmid (provided by Dr. Bernard Moss - NIH-NIAID) and confirmed by sequencing. The pLW44 transfer plasmid contains the green fluorescent protein coding sequence under the control of the mH5 early/late Orthopoxvirus vaccinia (VACV) promoter. MVA-RecBoAHV was produced using homologous recombination from infection (MVA) and transfection (pLW44-BoAHV) in a eucaryotic cell.
Comments: Figure 5 – 1) Standard deviation is not shown on the charts, why? Have you done statistical processing of the data? 2) figure’s legend is too large;
Response: As informed in the figure 5 legend, the animals were immunized in a prime-boost-boost protocol, and the samples collected after prime and boost 1 were evaluated in pool format and tested in duplicate, then no statistic was done. For samples collected after boost 2 that were individually assessed, it was performed using ANOVA multiple comparisons test with post-hoc Tukey’s test.
We agreed about the legend and it was reformulated. p-values have been added to Figure 5.
Figure 5. Evaluation of antibody responses in C57BL/6 mice after vaccination. Indirect ELISA anti-bovine IgG and IgM for BoAHV1 and BoAHV5 antigens from mice immunized with MVA-RecBoAHV. The animals were immunized in a prime-boost-boost protocol. All groups evaluated were tested on six animals per group. Prime and Boost 1 were evaluated in a pool format, being tested in two technical replicates. Boost 2 was evaluated individually in single replicates. The statistics presented for the boost 2 results correspond to: IgG BoAHV1 was performed using ANOVA multiple comparisons test with post-hoc Tukey’s. IgG BoAHV5 was performed using Kruskal–Wallis multiple comparisons test with post-hoc Dunn’s. IgM BoAHV5 was performed using ANOVA multiple comparisons test with post-hoc Tukey’s test. (lines 444-453)
Comments: L495 – “Mycobacterium tuberculosis” – should be in italic as species name;
Response: Thanks for the correction. Italics have been added.
Comments: References section – all references should be given in equal style; please check carefully and correct.
Response: We thank you for this comment. The references section has been carefully checked and formatted to ensure consistency throughout the manuscript.
Round 2
Reviewer 1 Report
Comments and Suggestions for Authors
No further comment.
Author Response
We thank you for your valuable contributions.
Reviewer 3 Report
Comments and Suggestions for Authors
Dear authors,
Thank you for the revised version of the manuscript, you did a lot of work to make it better.
However, the manuscript is still written somewhat carelessly and many issues still need correction.
General comments – 1) the names of the various sections/subparts of the manuscript must be given in a uniform style. Please correct; 2) you have used a lot of software and online services, the appropriate references are needed;
Authors section - Please check your punctuation;
Affiliation section – 1) Please check your punctuation, remove any extra periods, etc. 2) Affiliation 2 – the information about country should be added;
L18 – “Bovine” – no need to start with capital letter, correct, please;
L98-99 – “…CEUA/UFMG … UFMG-CEUA…” - please provide the same name(abbreviation) of the organization/committee;
L99 – “IACUC” – what is it? Should be stated;
L104 – “3Rs principle” – what is it? The appropriate reference (or explanation) should be provided;
L128 – “in-silico” – should be changed to “in silico” and given in italic;
L159 – “MVA-RecBoAHV viral vector was selected” – what is the vector? What manufacturer? Should be specified;
Section 2.5 - Please rewrite it in accordance with the logic of the experiment - first you got a vector from NIH, then you did the next steps…;
L175 – “with primers annealing…” – the reference to primers or primers’ sequences should be provided;
L176 – “using Chromas v2.23” – please change to “using Chromas v2.23 software”;
L177 – “using BLASTn” – the reference (if it was online service) or software package should be stated here;
L203 – “VACV WR” – what is it? Should be specified for the first time in text;
L217-218 – “UCFS” and “University of California San Francisco” – is it the same? If so, is the abbreviation correct?
L219 – “ul” – in other parts of text you used “μl” – please check and correct throughout the text;
L261-262 – “highlighted” – where? In Figure?
Table 2 – what do the underlined sequences mean?
L295 – “BepiPred 3.0” – software of online source? Should be indicated;
L305 – “NetMHCpan tool” – if this is online tool, the reference should be provided;
L366 – “E. coli” – should be in italic;
L367-368 – “molecular weight marker” – the manufacturer should be indicated;
Figure 2A – 1) it is not very clear where the target protein is located in the induced fraction, lanes I and UI are almost the same and the target protein of 12.7 kDa is not visible; the only thing that can be noticed is that less protein is applied in the UI fraction; 2) why fractions E1-E5 contain at least two bands?
Figure 2B – where is the target band detected with specific antibodies here? No 12.7 kDa band is visible on blots but a lot of other proteins;
Figure 2C – why IgG reacted worse with recombinant protein? Any idea? Should be discussed;
Figure 3 – it is unclear why the Boost 1 with recombinant protein resulted in extreme low levels of antibodies (even in comparison with Prime immunization). How could you explain that?
L407-408 – “were generated after recombination between the MVA genome and the pLW44 transfer plasmid” – what method/kit was used for this? Should be described in M&M section (L169-172);
Figure 5 – why the statistical analysis was performed only for Boost 2 evaluation, but not for Prime and Boost 1?
L476-491 – “HO”, “homologous protocol”, “HE”, “heterologous protocol” should be clearly described;
Table 3 – what do the blanks mean? Should be specified;
Author Response
Dear authors,
Thank you for the revised version of the manuscript, you did a lot of work to make it better.
However, the manuscript is still written somewhat carelessly and many issues still need correction.
Response: We thank the reviewer for the overall assessment and for recognizing our efforts to improve the manuscript. We carefully revised the entire text again to address language and formatting issues, ensuring clarity and consistency throughout the manuscript. We also incorporated the specific corrections suggested in the following comments.
General comments:
Comments: 1) the names of the various sections/subparts of the manuscript must be given in a uniform style. Please correct;
Response: We thank the reviewer for this observation. We carefully revised the manuscript to ensure that all section and subsection titles follow the uniform formatting style required by the journal’s guidelines. In addition, extra spaces between the section numbering and titles were removed to maintain consistency throughout the text.
Comments: 2) you have used a lot of software and online services, the appropriate references are needed;
Response: We thank the reviewer for this observation. We have carefully added appropriate references for all software and online tools used in the study.
Comments: Authors section - Please check your punctuation;
Response: We checked the punctuation as per the guideline, thank you.
Aline Aparecida Silva Barbosa 1,#, Samille Henriques Pereira 1,#, Mateus Laguardia-Nascimento 1, Amanda Borges Ferrari 1, Laura Jorge Cox 1, Raissa Prado Rocha 2, Victor Augusto Teixeira Leocádio 1, Ágata Lopes Ribeiro 1, Karine Lima Lourenço 3, Flávio Guimarães Da Fonseca 1 and Edel F. Barbosa-Stancioli 1,* (lines 97-98)
Comments: Affiliation section – 1) Please check your punctuation, remove any extra periods, etc. 2) Affiliation 2 – the information about country should be added;
Response: We checked the punctuation and added the country to affiliation 2.
School of Biosciences, University of Surrey. Guilford, Surrey, England. GU27XH; raissa.biotec@gmail.com (line 12)
Comments: L18 – “Bovine” – no need to start with capital letter, correct, please;
Response: Thank you for raising this question. In fact, Bovine in the L18 is the name of the species, and then, was written in capital letters. And according to modifications made by the Committee at ICTV in 2022, the name of the family and the name of the species had been changed. According to this fact, the text was modified as bellow:
Abstract - modified from: Bovine alphaherpesvirus 1 and Bovine alphaherpesvirus 5
to: Varicellovirus bovinealpha1 and Varicellovirus bovinealpha5 (line 18)
Keywords - modified from Bovine alphaherpesvirus 1 and Bovine alphaherpesvirus 5 ; BoAHV-1, BoAHV-5; Recombinant vaccine; Multi-epitope antigen; MVA vector.
To: Varicellovirus bovinealpha1 and Varicellovirus bovinealpha5 ; BoAHV-1, BoAHV-5; Recombinant vaccine; Multi-epitope antigen; MVA vector. (lines 43-44)
Introduction - modified from: Bovine alphaherpesvirus 1 (infectious bovine rhinotracheitis virus; BoAHV-1) and Bovine alphaherpesvirus 5 (bovine encephalitis herpesvirus; BoAHV-5) are two closely related species members of the Herpesviridae family, Alphaherpesvirinae subfamily, species Varicellovirus bovinealpha1 (infectious bovine rhinotracheitis virus; BoAHV-1) and Varicellovirus bovinealpha1 [2,3].
To: Varicellovirus bovinealpha1 (previously Bovine alphaherspervirus 1 / infectious bovine rhinotracheitis virus; BoAHV-1) and Varicellovirus bovinealpha5 (previously Bovine alphaherspervirus 5 / bovine encephalitis herpesvirus; BoAHV-5) are two closely related members of the Orthoherpesviridae family, Alphaherpesvirinae subfamily [2,3]. (lines 48-51)
Comments: L98-99 – “…CEUA/UFMG … UFMG-CEUA…” - please provide the same name(abbreviation) of the organization/committee;
Response: The abbreviation has been corrected, thanks for the observation.
The CEUA/UFMG follows all Institutional Animal Care and Use Committee (IACUC) guidelines to minimize animal suffering. (lines 96-97)
Comments: L99 – “IACUC” – what is it? Should be stated;
Response: Thank you for your observation, the name was inserted before the abbreviation.
The CEUA/UFMG follows all Institutional Animal Care and Use Committee (IACUC) guidelines to minimize animal suffering. (lines 97-98)
Comments: L104 – “3Rs principle” – what is it? The appropriate reference (or explanation) should be provided;
Response: The reference regarding the 3Rs principle has been added to the manuscript.
These sample sizes balance statistical sensitivity with the 3Rs principle [21] of reduction and the stepwise approach from pilot testing to larger confirmatory trials. (lines 102-104)
Russell WMS, Burch RL. The principles of humane experimental technique. London: Methuen; 1959.
Comments: L128 – “in-silico” – should be changed to “in silico” and given in italic;
Response: We have corrected it. It was an oversight on our part, caused by an automatic correction made by the computer.
The in silico prediction of T-cell and B-cell epitopes was performed [...] (line 128)
Comments: L159 – “MVA-RecBoAHV viral vector was selected” – what is the vector? What manufacturer? Should be specified;
Response: This is a viral vector that was contributed by our group. Vector construction was described in section 2.5 Construction of the Recombinant MVA-RecBoAHV Vector (lines 160-189)
Comments: Section 2.5 - Please rewrite it in accordance with the logic of the experiment - first you got a vector from NIH, then you did the next steps…;
Response: Thanks for the suggestion. The paragraph has been rewritten for better understanding.
The MVA used in these constructions is derived from the MVA-1974 clone, kindly provided by Dr. Bernard Moss (LVD/NIAID/NIH). This low-passaged MVA was transferred to UFMG under a specific material transfer agreement (MTA).The DNA fragment encoding the RecBoAHV multi-epitope protein initially contained in the pGEMT-easy vector was restricted with BamHI and HindIII enzymes (Promega, USA). The fragment was inserted into the pLW44 transfer plasmid using T/A cloning and confirmed by sequencing (see Figure S-1 for plasmid constructions). The pLW44 transfer plasmid contains the green fluorescent protein (GFP) coding sequence under the control of the mH5 early/late Orthopoxvirus vaccinia (VACV) promoter [26]. BHK-21 cells (Baby Hamster Kidney fibroblasts, obtained from the American Type Culture Collection - ATCC® CCL-10™) were infected with MVA and subsequently transfected with pLW44-RecBoAHV using Lipofectamine 3000 Reagent (Invitrogen, USA). The construction of the recombinant vaccine vector was based on the homologous recombination between the plasmid (pLW44-RecBoAHV) and MVA. Cells infected with recombinant vector clones were selected with the aid of GFP expression, and clones expressing MVA-RecBoAHV were subsequently isolated. The MVA-RecBoAHV construct was validated by Sanger sequencing using the BigDye® Terminator v1.1 Cycle Sequencing Kit (Applied Biosystems, USA) on a MegaBACE™ 1000 capillary sequencer (GE Healthcare, UK) with primers annealing to the pLW44 transfer plasmid flanking regions (5’ AAAGACCCCAACGAGAAGC 3’ and 5’ GTCTGAGGAAAAGGTGTAGCG 3’). Sequence data were analyzed using Chromas v2.23 software (https://technelysium.com.au) and aligned with reference sequences in GenBank using BLASTn (NCBI, USA - https://blast.ncbi.nlm.nih.gov/Blast.cgi). Transcript expression in infected BHK-21 cells was detected by RT-PCR using oligo(dT) and BoAHV-specific primers (5’ ATGTCGACCACCGCGAGCACACC 3’ and 5’ AGCTGCAGGTCGCTGCTATCGCC 3’). For the viral stock production, cells were infected at 37ºC, 5% CO2 atmosphere for 1 hour. After adsorption, the cells were incubated in the same atmosphere in DMEM (Dulbecco's Modified Eagle Medium – Sigma Aldrich, USA) supplemented with 7.5% NaHCO3, antibiotics (100 μg/mL streptomycin and 100 U/mL penicillin), antifungal (fungizone at 25 μg/mL) and 5% fetal bovine serum (FBS - Gibco, USA) for 48 hours. Recombinant virus purification was performed in 36% (w/v) sucrose cushion (in 10 mM Tris-HCl, pH 9.0), and the titer was determined by a plaque-based assay [27,28]. (lines 161-192)
Comments: L175 – “with primers annealing…” – the reference to primers or primers’ sequences should be provided;
Response: Thanks for the note. We've added the primer information.
[...] with primers annealing to the pLW44 transfer plasmid flanking regions (5’ AAAGACCCCAACGAGAAGC 3’ and 5’ GTCTGAGGAAAAGGTGTAGCG 3’). (lines 179-180)
Transcript expression in infected BHK-21 cells was detected by RT-PCR using oligo(dT) and BoAHV-specific primers (5’ ATGTCGACCACCGCGAGCACACC 3’ and 5’ AGCTGCAGGTCGCTGCTATCGCC 3’. (lines 184-185)
Comments: L176 – “using Chromas v2.23” – please change to “using Chromas v2.23 software”;
Response: Thank you for your observation. It has been added to the manuscript.
Sequence data were analyzed using Chromas v2.23 software (https://technelysium.com.au) [...] (lines 180-181)
Comments: L177 – “using BLASTn” – the reference (if it was online service) or software package should be stated here;
Response: Thank you for your observation. It has been added to the manuscript.
[...] with reference sequences in GenBank using BLASTn (NCBI, USA - https://blast.ncbi.nlm.nih.gov/Blast.cgi). (line 182)
Comments: L203 – “VACV WR” – what is it? Should be specified for the first time in text;
Response: WR is the strain of the virus (Western Reverse). This information was missing from the manuscript and has been added. Thank you.
[...] inactivated VACV strain Western Reserve (WR). (lines 210-211)
Comments: L217-218 – “UCFS” and “University of California San Francisco” – is it the same? If so, is the abbreviation correct?
Response: The abbreviation was misspelled. Thanks for the correction.
According to the UCSF Office of Research [...] (lines 224)
Comments: L219 – “ul” – in other parts of text you used “μl” – please check and correct throughout the text;
Response: Thank you for your observation. The correction was made to the manuscript.
[...] the dose recommended for intranasal instillation of mice is 50 µl maximum volume, and following this recommendation, we prepared individual doses of 10 µl (divided equally between both nostrils). (lines 225-227)
Comments: L261-262 – “highlighted” – where? In Figure?
Response: We thank the reviewer for the observation. The information in question refers to the results presented in Table 3 and should not have been included in the Methods section. Accordingly, it has been removed from the Materials and Methods and added only to the legend of Table 3 (lines 505-507).
Comments: Table 2 – what do the underlined sequences mean?
Response: The underlined sequences refer to the linkers adding the final RecBoAHV sequence. The information has been added to the end of the table.
The underlined sequences refer to the linkers adding the final RecBoAHV sequence. (line 298)
Comments: L295 – “BepiPred 3.0” – software of online source? Should be indicated;
Response: Thank you for the observation. The information has been added to the manuscript at the first mention of the tool.
[...] the BepiPred-3.0 tool online source (https://services.healthtech.dtu.dk/services/BepiPred-3.0/) [25] was employed [...] (lines 138-139)
Comments: L305 – “NetMHCpan tool” – if this is online tool, the reference should be provided;
Response: Thank you for the observation. The information has been added to the manuscript at the first mention of the tool.
[...] were carried out using the NetMHCpan 4.1 tool (DTU Health Tech - Bioinformatic Services, https://services.healthtech.dtu.dk/services/NetMHCpan-4.1/) (lines 131-132)
Comments: L366 – “E. coli” – should be in italic;
Response: We corrected ourselves, thank you.
Comments: L367-368 – “molecular weight marker” – the manufacturer should be indicated;
Response: The manufacturer has been added to the manuscript, thank you.
(M) molecular weight marker (Sigma-Aldrich, EUA) (lines 376-377)
Comments: Figure 2A – 1) it is not very clear where the target protein is located in the induced fraction, lanes I and UI are almost the same and the target protein of 12.7 kDa is not visible; the only thing that can be noticed is that less protein is applied in the UI fraction; 2) why fractions E1-E5 contain at least two bands?
Response:
- We thank you for this observation. The target protein can indeed be observed in both induced (I) and uninduced (UI) fractions. This is due to basal (“leaky”) expression of the recombinant protein in the uninduced condition, which is a common phenomenon in bacterial expression systems and has been reported previously in the literature. Although the amount of protein in the UI fraction is lower, its presence does not interfere with downstream analyses. We have added this clarification to the Results section.
- The presence of two bands in fractions E1–E5 is due to the characteristic migration pattern of the target protein, which can appear as multiple bands in SDS-PAGE. This may result from minor degradation, the presence of the initiating methionine, or different conformational states of the recombinant protein. Such behavior is commonly observed for proteins of similar size and does not affect subsequent analyses. We have added this clarification to the Results section.
The RecBoAHV coding gene was subcloned into the pQE30 plasmid and used to transform E. coli M15 cells. The transformant bacteria were cultured and induced with IPTG to drive expression of the recombinant protein, and the expression was evaluated by SDS-PAGE (Figure 2A). As predicted, a recombinant multi-epitope protein of approximately 12.7 kDa was detected. A small amount of the protein was also observed in the uninduced fraction, due to basal (“leaky”) expression, which is common in bacterial expression systems. The protein was efficiently purified using affinity chromatography on a Ni-NTA chelating column (Figure 2A), and fractions E1–E5 displayed a characteristic pattern of two bands, likely resulting from minor degradation, the presence of the initiating methionine, or different conformational states of the recombinant protein. (lines 352-361)
Comments: Figure 2B – where is the target band detected with specific antibodies here? No 12.7 kDa band is visible on blots but a lot of other proteins;
Response: Figure 2B is a figure composed by three different western blots using BoAHV-1 (purified virions), BoAHV-5 (purified virions), and RecBoAHV, from different experiments. Therefore, we prepared Figure 2 with the size marker used in all three experiments (the precision plus protein TM Kaleidoscope TM Prestained Protein Standards #1610375 - BIO-RAD), but inserting a figure from the Bio-Rad site. The marker figure was placed slightly below the largest proteins seen on the gel. If it were repositioned (which we believe should not be altered at this time), the recombinant protein labeling would be between the sizes of 20 to 12 kDa. Nonetheless, the original blots were sent to demonstrate that size markers are indeed correct and correspond to the size marker in the figure.
In Figure 2A we can see that the purified recombinant protein shows a run on the polyacrylamide gel varying around 22 to 12 kDa in the different eluates, and the specific sera recognize this “pattern”. The multi-epitope protein was designed with the flexible ligand G-S-G-S-G among the different epitopes, so epitopes could be presented in a way that makes them more easily recognizable. Literature has shown that the use of flexible linkers alters the folding of recombinant proteins, also changing their migration in polyacrylamide gels, potentially displaying different size patterns compared to the molecular marker, yet being the same protein, and capable of being recognized by specific sera.
Comments: Figure 2C – why IgG reacted worse with recombinant protein? Any idea? Should be discussed;
Response: We thank you for this comment. When using total antigen, a broad range of antibodies can be recognized, whereas the recombinant protein only contains specific epitopes, and therefore recognizes only antibodies directed against these epitopes. This explains the lower IgG reactivity observed with the recombinant protein, and this phenomenon was expected based on the design of the multi-epitope construct. We have added this discussion to the Results section.
As expected, IgG reactivity was lower with the recombinant protein compared to total antigen, since the recombinant protein contains only selected epitopes and thus recognizes a narrower subset of antibodies. (lines 369-371)
Comments: Figure 3 – it is unclear why the Boost 1 with recombinant protein resulted in extreme low levels of antibodies (even in comparison with Prime immunization). How could you explain that?
Response: Unfortunately, we do not have a technical explanation for that particular data. Despite having repeated the experiment a few times, ODs in the ELISA after the first boost were consistently low. We prefer to refrain from giving any possible explanation, as it would be simply speculative. Nonetheless, results after the second boost showed that the recombinant protein is clearly immunogenic.
Comments: L407-408 – “were generated after recombination between the MVA genome and the pLW44 transfer plasmid” – what method/kit was used for this? Should be described in M&M section (L169-172);
Response: Supplementary information has been added to section 2.5.
BHK-21 cells (Baby Hamster Kidney fibroblasts, obtained from the American Type Culture Collection - ATCC® CCL-10™) were infected with MVA and subsequently transfected with pLW44-RecBoAHV using Lipofectamine 3000 Reagent (Invitrogen, USA). The construction of the recombinant vaccine vector was based on the homologous recombination between the plasmid (pLW44-RecBoAHV) and MVA. (lines 169-174)
Comments: Figure 5 – why the statistical analysis was performed only for Boost 2 evaluation, but not for Prime and Boost 1?
Response: We thank you for this observation. Serum collection from mice after the Prime and Boost 1 immunizations yielded only small volumes, as collections were performed without euthanizing the animals. Due to the limited serum, samples from these time points were tested in pooled format. After Boost 2, the animals were euthanized, allowing collection of larger serum volumes and enabling individual testing. Consequently, only two data points were available for Prime and Boost 1, which did not allow for statistical analysis. Statistical evaluation was therefore performed only for the Boost 2 time point. The test format was described in the caption of Figure 5. (lines 459-461)
Comments: L476-491 – “HO”, “homologous protocol”, “HE”, “heterologous protocol” should be clearly described;
Response: Clearer information about abbreviations has been added to the text. Thank you for your observation.
Comparative analysis of ELISA indices for IgG and IgM revealed distinct responses across the animal models and immunization regimens. In rabbits immunized with RecBoAHV, robust IgG responses were observed against BoAHV-1 and BoAHV-5, with indices reaching 7.89 and 6.22, respectively, after the second boost. In contrast, rabbits vaccinated with MVA-RecBoAHV HO (homologous protocol – prime and boosts with MVA-RecBoAHV) exhibited more moderate responses, with indices of 2.94 against BoAHV-1 and 5.39 against BoAHV-5 following the second boost.
In the mice experiments, all immunizing agents induced seroconversion for both IgG and IgM, with indices above the cutoff as early as the priming dose. Animals immunized with RecBoAHV exhibited a progressive increase in IgG and IgM indices against both viruses, reaching up to 1.75 (IgG anti-BoAHV-5) and 1.72 (IgM anti-BoAHV-1) after the second boost. Vaccination with MVA-RecBoAHV HO induced moderate responses, with IgG indices ranging from 1.11 to 1.85 and IgM indices from 1.19 to 1.57. MVA-RecBoAHV HE (heterologous protocol – prime with RecBoAHV and boosts with MVA-RecBoAHV) elicited the highest serological responses, particularly for IgG anti-BoAHV-5 (3.90) and IgM anti-BoAHV-5 (4.80) after the first boost. The inactivated BoAHV-1/5 vaccine promoted intermediate levels of IgG (up to 2.00) and IgM (up to 2.16) against both viruses. (lines 485-501)
Comments: Table 3 – what do the blanks mean? Should be specified;
Response: Blank spaces have been replaced with (-) and an explanation has been added to the legend.
(-) indicates that experiments were not performed for these points. (line 508)
Round 3
Reviewer 3 Report
Comments and Suggestions for Authors
Dear authors,
Thank you for your revised manuscript, you did a lot of work and now it looks really great. All my issues were solved or explained, thank you.
Only one little remark - please check and correct som references in Reference section - some doi: information is underlined.
Author Response
Comments:
Dear authors,
Thank you for your revised manuscript, you did a lot of work and now it looks really great. All my issues were solved or explained, thank you.
Only one little remark - please check and correct som references in Reference section - some doi: information is underlined.
Response:
Dear reviewer,
We sincerely thank you for the positive evaluation and final remark. The references were carefully checked, and all underlined DOIs have been corrected to ensure consistency in formatting.